# Mixing between chemically variable primitive basalts creates and modifies crystal cargoes

David A. Neave [1,2✉], Philipp Beckmann[2], Harald Behrens[2] & François Holtz[2]

Basaltic crystal cargoes often preserve records of mantle-derived chemical variability that have been erased from their carrier liquids by magma mixing. However, the consequences of mixing between similarly primitive but otherwise chemically variable magmas remain poorly understood despite ubiquitous evidence of chemical variability in primary melt compositions and mixing-induced disequilibrium within erupted crystal cargoes. Here we report observations from magma–magma reaction experiments performed on analogues of primitive Icelandic lavas derived from distinct mantle sources to determine how their crystal cargoes respond to mixing-induced chemical disequilibrium. Chemical variability in our experimental products is controlled dominantly by major element diffusion in the melt that alters phase equilibria and triggers plagioclase resorption within regions that were initially plagioclase saturated. Isothermal mixing between chemically variable basaltic magmas may therefore play important but previously underappreciated roles in creating and modifying crystal cargoes by unlocking plagioclase-rich mushes and driving resorption, (re-)crystallisation and solid-state diffusion.

[1] Department of Earth and Environmental Sciences, The University of Manchester, Manchester, UK. [2] Leibniz Universität Hannover, Institut für Mineralogie, Hannover, Germany. ✉email: david.neave@manchester.ac.uk

Chemical variability in primitive mid-ocean ridge and ocean island basalts (MORBs and OIBs, respectively; oceanic basalts, collectively) results from variability in mantle melting processes and source compositions[1,2]. Correlations between the isotopic and incompatible-element compositions of erupted basalts reflect how the subduction and subsequent in-mixing of oceanic lithosphere by mantle convection have created deep isotopic and chemical heterogeneities through geological time[3]. Subduction has also created lithological heterogeneities that propagate into correlated variations in the major and trace element compositions of erupted basalts[4,5]. Thus, similarly primitive but otherwise chemically variable oceanic basalts have different phase equilibria at the same pressure–temperature ($P$–$T$) conditions and consequently evolve along compositionally distinct trajectories[6,7]. However, mantle-derived chemical variability is progressively erased by mixing during magma ascent and evolution[8,9], meaning that erupted basalts are typically less diverse than the primary melts from which they have evolved[10]. Fortunately, vital records of mantle-derived chemical variability are often preserved within crystals and melt inclusions that are relatively more resistant to mixing-induced re-equilibration than their carrier liquids[11–15]. Nevertheless, surprisingly little is known about the nature of crystal-melt interactions taking place during mixing between chemically variable oceanic basalts and their roles in creating and modifying crystal cargoes.

Records of mantle-derived variability in basaltic crystal cargoes are associated with disequilibrium features that range from simple normal zoning to complex textures reflecting resorption, rapid crystal growth, and diffusive re-equilibration[16–18]. Crystal-hosted melt inclusions provide complementary archives that confirm the presence of mantle-derived chemical variability within individual magma plumbing systems[13,14]. Indeed, this chemical variability may play a fundamental role in creating some types of melt inclusion[19,20]. Disequilibrium features in crystal cargoes record changes in magma $P$–$T$–$H_2O$ activity ($a_{H_2O}$)–oxygen fugacity ($f_{O_2}$)–composition ($X$) conditions as well as the timescales over which these changes occur. For example, compositional zoning in plagioclase crystals can preserve information about magma reservoir processes and transport pathways[21–23], while compositional zoning in olivine crystals often records timescales of magma storage and ascent[24–26]. Although recharge by hot and primitive magmas is a long-recognised mechanism for creating disequilibrium features within crystal cargoes[17,18,27], near-isothermal mixing between chemically variable magmas is also likely to have important but as yet ill-defined impacts.

Experimental simulations of magma mixing typically focus on physical mingling in dynamic experiments[28–32] or diffusive re-equilibration in classic melt–melt couples[33,34]. While physical mingling is an essential component of magma mixing, diffusion over short (i.e. µm) lengthscales ultimately changes local melt compositions and leads to the modification of crystals by resorption, (re-)crystallisation and solid-state diffusion. However, melt–melt couple experiments are, by definition, performed under superliquidus conditions and thus provide limited insights into magma–magma reactions[33,35]. Although constitutional undercooling and diffusive controls over crystal growth and resorption have been investigated in some simple systems[36–40], and crystal-bearing mixing experiments have been performed in the contexts of enclave formation and andesite genesis by magma hybridisation[32,41], current observations typically constrain how pressure and temperature affect mineral stabilities rather than melt composition.

Mantle-derived chemical variability is especially well characterised in basalts from southwest Iceland. At any given MgO content, incompatible element-depleted basalts from lherzolitic mantle sources are rich in $Al_2O_3$ and CaO but poor in FeO* (total Fe expressed as FeO) and $Na_2O$, while incompatible element-enriched basalts from recycled mantle sources are poor in $Al_2O_3$ and CaO but rich in FeO* and $Na_2O$[42]. This dichotomy is exemplified by the Háleyjabunga and Stapafell lavas on the Reykjanes Peninsula (Fig. 1a, b)[13]. Crystallisation experiments performed on synthetic analogues of these lavas illustrate how mantle-derived variability in major element compositions affects their phase equilibria[7]. Namely, plagioclase crystallises at >1200 °C from the $Al_2O_3$-rich and incompatible element-depleted Háleyjabunga lava analogue but only at <1180 °C from the $Al_2O_3$-poor and incompatible element-enriched Stapafell lava analogue (Fig. 1c, d). For any given decrease in temperature, incompatible element-depleted magmas thus crystallise a greater proportion of their mass than their incompatible element-enriched counterparts, potentially resulting in the deep sequestration of incompatible element-depleted magmas as plagioclase-rich cumulates and the progressive biasing of evolved magmas towards incompatible element-enriched compositions[7].

Here we present the results of magma–magma reaction experiments designed to determine how mixing-induced chemical disequilibrium affects the crystal cargoes of chemically variable oceanic basalts. These experiments were performed by juxtaposing synthetic magma analogues of the Háleyjabunga and Stapafell lavas at 300 MPa and 1190 °C for durations of 1–96 h under realistic magma storage conditions: 300 MPa, 1190 °C, low water activities (~0.06) and oxygen fugacities approximately one log unit above the fayalite-magnetite-quartz buffer (full details are provided in the methods)[7,43]. In this contribution, we first examine phase relations across juxtaposed magma analogues and describe how crystals responded to the diffusive re-equilibration of melt major element contents between initially distinct magmas. We then explore the implications of our findings for the creation and modification of basaltic crystal cargoes by magma mixing and mush disaggregation.

## Results

**Magma synthesis experiments**. The products of synthesis experiments performed on the incompatible element-depleted Háleyjabunga lava analogue and incompatible element-enriched Stapafell lava analogue contain crystals of olivine, clinopyroxene and plagioclase (Fig. 2a, c and Supplementary Data 1, 2), and olivine and clinopyroxene (Fig. 2b, d), respectively. Although the products of the experiment on the Stapafell analogue have a lower glass content (i.e. melt mass fraction, $F$) than those of the experiment on the Háleyjabunga analogue ($F = 0.72$ and 0.90, respectively, according to mass balance; Supplementary Data 7), glasses are interconnected throughout the products of both synthesis experiments. Crystals are typically euhedral to subhedral, though some plagioclase crystals are skeletal and contain melt inclusions. While clinopyroxene crystals sometimes form aggregates, degrees of crystal impingement are generally low. Some olivine crystals contain small inclusions of plagioclase (Fig. 2d).

**Magma–magma reaction experiments**. The products of magma–magma reaction experiments share textural and mineralogical characteristics with products of magma synthesis experiments (Figs. 3 and 4 and Supplementary Data 3–6). Namely, crystals are smaller and more abundant in products derived from the incompatible element-depleted Háleyjabunga analogue than those from the incompatible element-enriched Stapafell analogue. Overall, textures of crystals far from original interfaces do not appear to evolve with increasing experimental

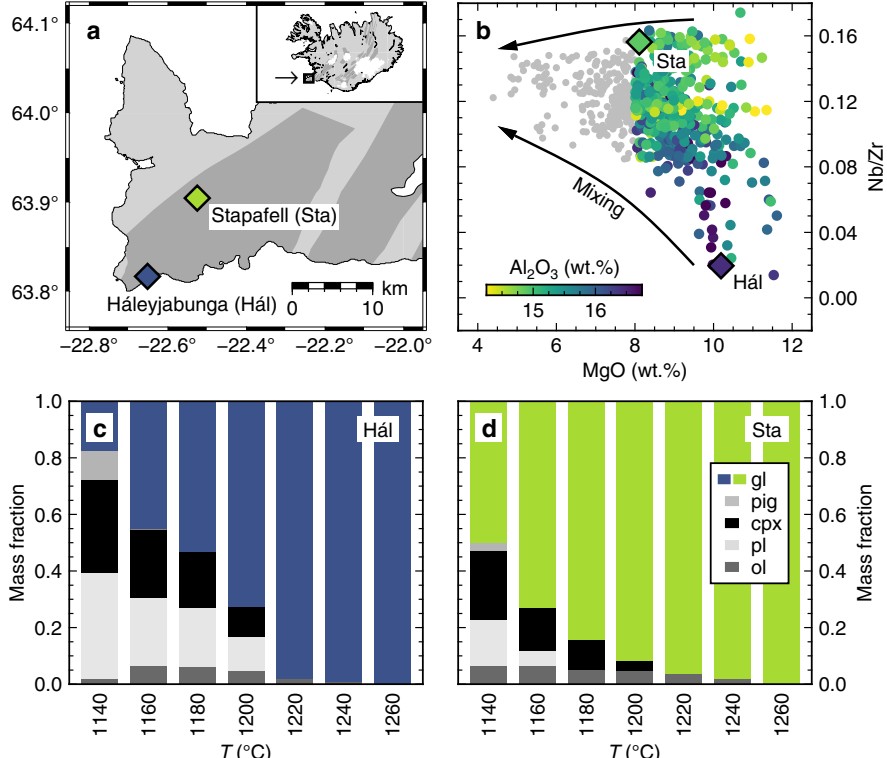

**Fig. 1 Mantle-derived chemical variability in Icelandic basalts and its effects on the phase equilibria of primitive magmas. a** Locations of the incompatible element-depleted Háleyjabunga (Hál) and incompatible element-enriched Stapafell (Sta) lavas on the Reykjanes Peninsula of southwest Iceland. Volcanic systems are highlighted in dark grey. This map (and all subsequent plots) were produced with The Generic Mapping Tools (GMT) 6[80]. **b** Summary of major and trace element variability in lavas from southwest Iceland[42]. MgO reflects the degree of magmatic evolution and Nb/Zr the degree of mantle-derived incompatible element enrichment. At any given MgO content, incompatible element-enriched lavas like those from Stapafell have lower $Al_2O_3$ contents than incompatible-element depleted lavas like those from Háleyjabunga. Arrows illustrate the reduction in chemical variability that results from concurrent mixing and crystallisation during magmatic evolution[8,9]. **c, d** Phase equilibria of Háleyjabunga (**c**) and Stapafell (**d**) lava analogues derived from crystallisation experiments at 300 MPa[7]. Equilibrium phase assemblages and proportions are related to the degree of incompatible element enrichment: plagioclase crystallises at higher temperatures and in greater amounts from incompatible element-depleted magmas that are correspondingly enriched in refractory elements like CaO and $Al_2O_3$. Phases are labelled as follows: gl glass, pig pigeonite, cpx clinopyroxene, pl plagioclase, ol olivine.

duration; only clinopyroxene crystals in portions of experimental products derived from the Stapafell analogue show possible evidence of Ostwald ripening (Fig. 4d). Although crystal fractions remain broadly constant with increasing experimental duration in far-field portions of experimental products derived from the Háleyjabunga analogue away from initial magma–magma interfaces (Figs. 3 and 4a, b), they appear to decrease in those derived from the Stapafell analogue (Figs. 3 and 4c, d). However, mass balance demonstrates that far-field melt mass fractions change little through time: between 1 and 96 h, F changes from 0.77 to 0.80 in far-field portions of the Háleyjabunga analogue, and from 0.92 to 0.93 in far-field portions of the Stapafell analogue. It is therefore likely that apparent differences in observed crystal fractions are related to heterogeneities created by gravitational settling during the synthesis experiment on the Stapafell analogue and that disequilibrium crystal fractions were retained through magma–magma reaction experiments. Somewhat lower crystal fractions in the products of magma–magma reaction experiments with respect to the products of synthesis experiments also reflect samples of the latter being sourced from capsule ends that lay slightly outside the vessel's hot zone.

Glass composition profiles through the products of magma–magma reaction experiments have variably linear and sigmoidal forms that depend on both experimental duration and the specific element in question (Fig. 5). Profiles near the central axes of experimental products have similar forms to those near

capsule walls, meaning that parallel profiles could be stacked on their common midpoints. Importantly, the similarity of parallel profiles demonstrates that advection was negligible once magma–magma interfaces were established (early advection is reflected in curved magma–magma interfaces in the products of 1- and 4-h experiments) and that experimental products record dominantly diffusive signals.

Far-field glasses derived from the Stapafell analogue have lower $Al_2O_3$ contents and higher FeO*, $TiO_2$ and $K_2O$ contents than those derived from the Háleyjabunga analogue after short experimental durations of 1 and 4 h (~14.5 versus ~16.0 wt.% for $Al_2O_3$, ~11.0 versus ~10.0 wt.% for FeO*, ~1.65 versus ~0.75 wt.% for $TiO_2$ and ~0.22 versus ~0.10 wt.% for $K_2O$, respectively; Fig. 5a–c, e). Differences in far-field $Al_2O_3$ and $TiO_2$ contents remain clear after longer experimental durations of 24 and 96 h while those in FeO* and $K_2O$ contents are muted or overprinted. Differences in CaO and $Na_2O$ contents are visible after 1 and 4 h (~12.0 versus ~12.6 wt.% for CaO and ~2.1 versus ~1.7 wt.% for $Na_2O$, respectively; Fig. 5d, f) but challenging to distinguish from analytical uncertainty after 24 and 96 h ($2\sigma = \pm 0.44$ wt.% and $\pm 1.3$ wt.% for CaO and $Na_2O$, respectively). Small differences in MgO contents (~8.3 versus ~8.5 wt.%) also cannot be observed after 4 h (Fig. 5g); the slightly higher MgO content of products from magma–magma reaction experiments with respect to synthesis experiments is consistent with the latter having equilibrated slightly outside the vessel's hot zone.

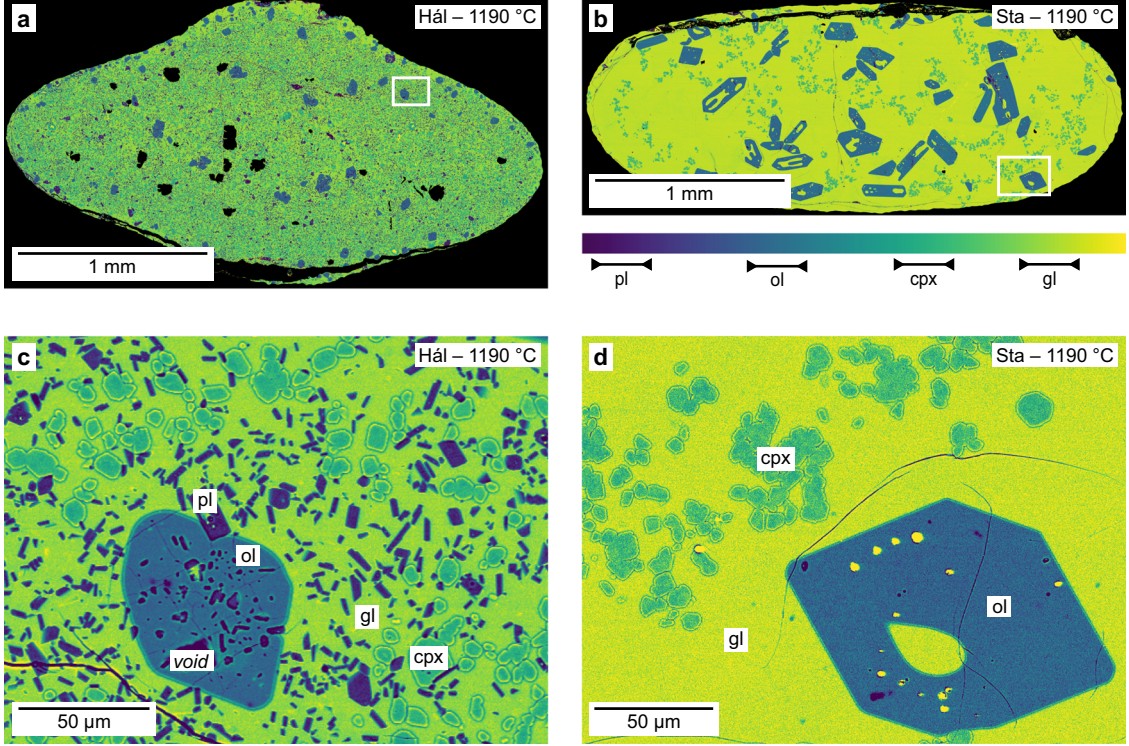

**Fig. 2 False-colour backscattered electron (BSE) images of the products of synthesis experiments.** Synthesis experiments on Háleyjabunga (Hál) and Stapafell (Sta) lava analogues were both performed at 300 MPa and 1190 °C. These conditions were chosen to maximise differences in equilibrium phase equilibria[7]. **a** Products of the synthesis experiment on the incompatible element-depleted Háleyjabunga lava analogue contain relatively modest amounts of glass [gl, yellow-green; melt mass fraction ($F$) ~ 0.72]. Black areas within the products are voids and cracks generated during polishing. **b** Products of the synthesis experiment on the incompatible element-enriched Stapafell lava analogue contain relatively large amounts of glass (gl; $F$ ~ 0.90). **c, d** High-resolution images of (**a**) and (**b**) showing crystals of olivine (ol, blue), clinopyroxene (cpx, blue-green) and plagioclase (pl, purple) in the products of the experiment on the Háleyjabunga analogue (**c**), and olivine and clinopyroxene crystals in the products of the experiments of the Stapafell analogue (**d**).

$Al_2O_3$ $TiO_2$ and $K_2O$ profiles have sigmoidal forms that show progressively more gradual transitions between far-field compositions with increasing experimental durations (Fig. 5a, c, e). FeO* profiles are sigmoidal in the products of 1- and 4-h experiments, but near-linear in the products of 24- and 96-h experiments (Fig. 5b). FeO* contents are also displaced to lower mean FeO* contents in the products of the 24- and 96-hexperiments with respect those of the 1- and 4-h experiments. Mass balance indicates that this results from modest Fe loss from $Au_{80}Pd_{20}$ capsules in the case of former (1–2% relative) and modest Fe gain in the case of the latter (2–5% relative)[44]. CaO and $Na_2O$ profiles are sigmoidal in the products of 1- and 4-h experiments, but linear in the products of 24- and 96-h experiments (Fig. 5d, f).

**Diffusive re-equilibration of experimental glasses.** Normalising distances through experimental products by the square root of experimental duration demonstrates that glass compositions are primarily controlled by diffusive re-equilibration between chemically distinct melts in the starting magma analogues[34,45]; $Al_2O$ and $TiO_2$ profiles are shown in Fig. 6. This is because diffusion distance scales with the square root of the diffusion coefficient multiplied by diffusion time (i.e. $x \propto \sqrt{Dt}$, where $x$ is the diffusion distance, $D$ is the diffusion coefficient and $t$ is time)[45], meaning that diffusively controlled composition profiles stack once time is cancelled out[34]. Overlapping profiles in Fig. 6 demonstrate that diffusion operated coherently throughout our magma–magma reaction experiments and that observations can be integrated across our different experiments.

Effective binary diffusion coefficients estimated by fitting error functions through glass composition profiles to solve Fick's 2$^{nd}$

Law are consistent with the modest body of published values available (Fig. 7)[35–38,46–49]. Most published diffusion coefficients for basaltic melts have been derived from crystal dissolution experiments[36–38] or classic melt–melt couples[35,46]. Na diffusivity has also been estimated from radiotracer diffusion experiments[47]. Where error functions could be fitted to our glass composition profiles by assuming that melt viscosity does not vary across experimental products (Supplementary Data 7), estimated effective binary diffusion coefficients are typically within one ln unit of regressions through published datasets that follow Arrhenian relationships ($D = D_0 e^{-E_A/RT}$, where $D$ is the diffusion coefficient, $D_0$ is the pre-exponential factor, $E_A$ is the activation energy for diffusion and $R$ is the gas constant; Fig. 7).

Diffusion coefficients estimated from our 1- and 4-h experiments are particularly close to global regressions for $Al_2O_3$, FeO* and CaO ($r^2 = 0.83$, 0.92 and 0.79 respectively), suggesting that the behaviour of these elements in basaltic melts can be largely explained without complex multicomponent diffusion models (Fig. 7a, b, d). While our estimated diffusion coefficients for $TiO_2$ are similar to published values, the latter span three ln units at any given $T$ and global regression statistics are modest ($r^2 = 0.59$), suggesting that $TiO_2$ may diffuse by a different mechanism from $Al_2O_3$, FeO* and CaO, at least when present a relatively low concentrations (Fig. 7c). Although published diffusion coefficients are scarce for $K_2O$ and $Na_2O$[46,47], our estimates fall on plausibly Arrhenian trends with the few data available (Fig. 7e, f). Estimated diffusivities of $Al_2O_3$ and $TiO_2$ are similar to theoretical Eyring diffusivities ($D_E = k_B T/\lambda\eta$, where $D_E$ is the Eyring diffusivity, $k_B$ is Boltzmann's constant, $\lambda$ is a jump distance of 0.4 nm related to the atomic spacing of silicate liquids

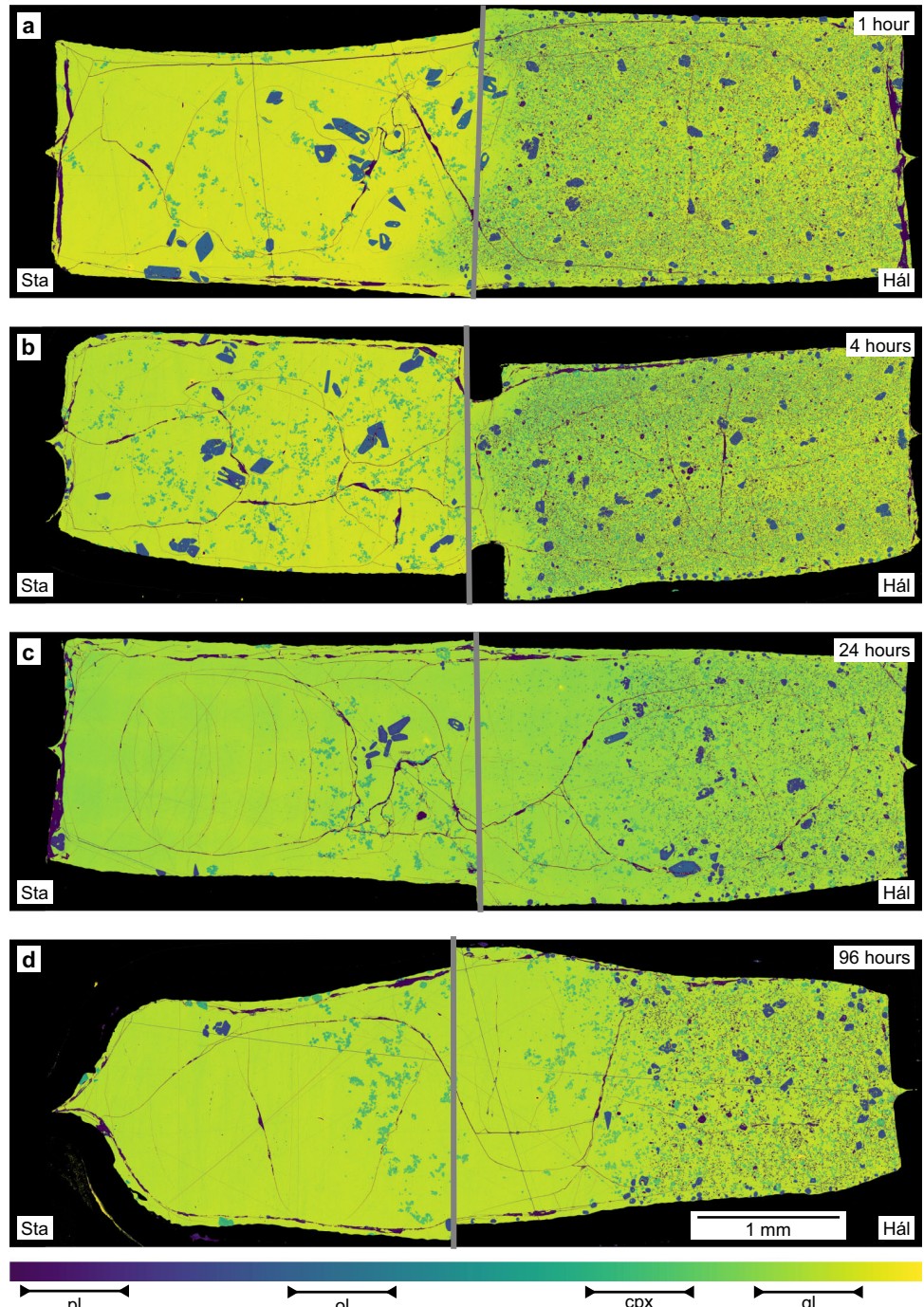

**Fig. 3 False-colour backscattered electron (BSE) maps of the products of magma–magma reaction experiments.** Magma–magma reaction experiments were performed by juxtaposing quenched magma cylinders from synthesis experiments on both Háleyjabunga (Hál) and Stapafell (Sta) lava analogues within new capsules. These new capsules were then subjected to exactly the same conditions used during synthesis experiments (300 MPa and 1190 °C). **a–d** BSE maps show the products of experiments run for 1 h (**a**), 4 h (**b**), 24 h (**c**) and 96 h (**d**). Magma cylinders synthesised from the incompatible element-depleted Háleyjabunga lava analogue are on the right, and magma cylinders synthesised from the incompatible element-enriched Stapafell lava analogue are on the left. Grey lines show the positions of original interfaces between juxtaposed magma cylinders. Plagioclase (pl) is the lowest BSE intensity phase (purple, only towards the right), followed, in order of increasing intensity, by olivine (ol, blue), clinopyroxene (cpx, blue-green) and glass (gl, green to yellow). Occasional bright flecks are small fragments of capsule material detached during sample preparation and are of no geological significance.

and $\eta$ is the average viscosity of the two end-member melts (12.5 Pa.s), calculated here with the model of Giordano et al.[50])[51]; Eyring diffusivities are typically good at describing the behaviour of relatively high-field strength network-forming cations (i.e. Si,

Al and Ti) and oxygen[34]. In contrast, estimated diffusivities of lower-field strength $Na_2O$ and $K_2O$ are higher, reflecting variable degrees of decoupling from network-forming cations (Fig. 7e, f)[52,53]. Finally, we note that melt $H_2O$ contents were not reported

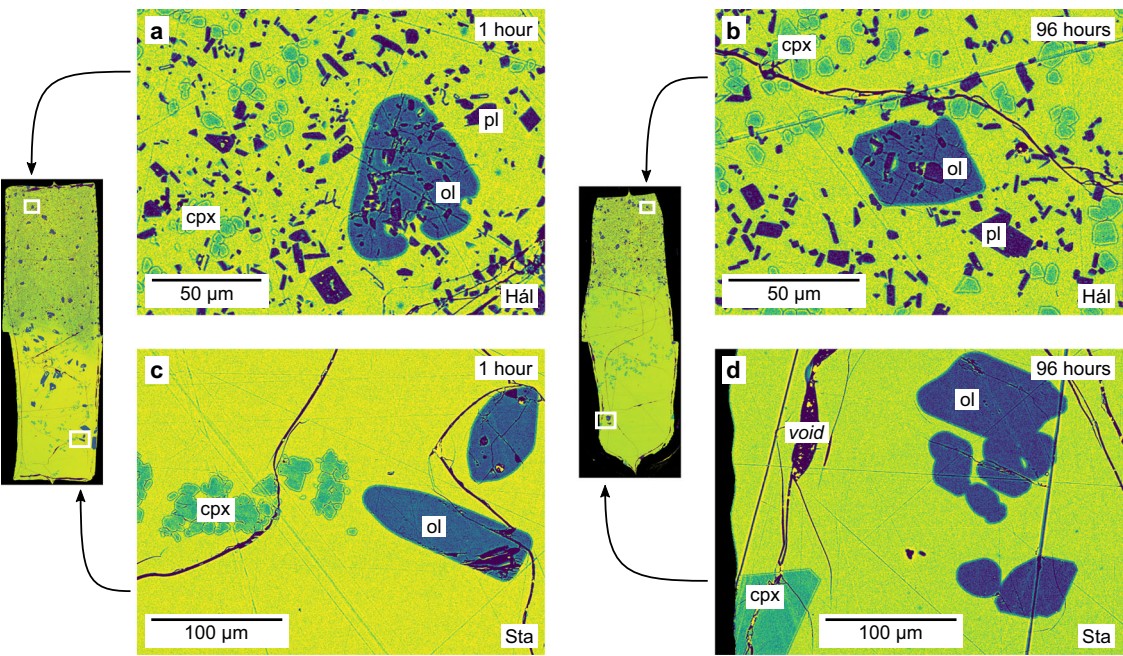

**Fig. 4 False-colour backscattered electron (BSE) images of the products of 1- and 96-h magma–magma reaction experiments. a, b** Neither the far-field texture nor abundance of crystals change appreciably in portions of the experimental products derived from the Háleyjabunga (Hál) lava analogue far from original magma–magma interfaces as experimental duration increases from 1 h (**a**) to 96 h (**b**). There is no clear evidence of Ostwald ripening in these portions of our experimental products. **c, d** Although the far-field abundance of crystals appears to decrease in portions of experimental products derived from the Stapafell (Sta) lava analogue far from original magma–magma interfaces as experimental duration increases from 1 h (**c**) to 96 h (**d**), crystal textures remain mostly the same. Specifically, while olivine crystals have similar sizes and textures after 1 and 96 h, clinopyroxene crystals may have decreased in number and increased in size by Ostwald ripening (i.e. coarsening). Phases are labelled as follows: gl glass (green to yellow), cpx clinopyroxene (blue-green), ol olivine (blue), pl plagioclase (purple).

in the published data considered here, and that small variations in melt $H_2O$ contents (~1 wt.%) affect the diffusivity of network-forming and high-field strength elements considerably[45].

**Mineral stabilities and compositions.** Olivine crystals occur in the products of magma synthesis experiments on both lava analogues (Figs. 2 and 8a). However, they have slightly higher forsterite contents [$X_{Fo}$ ~ 0.852 versus 0.843, respectively, where $X_{Fo} = Mg/(Mg + Fe)$ on a molar basis] in the products of the experiment on the FeO*-poor Háleyjabunga analogue than those of the experiment on the FeO*-rich Stapafell analogue. This difference is reflected in steps in mean $X_{Fo}$ across magma–magma interfaces in the products of 1- and 4-h experiments (Fig. 8a). Unfortunately, the abundance of olivine crystals is insufficient to determine whether these $X_{Fo}$ steps persist in the products of 24- and 96-h experiments. Some olivine crystals from longer-duration experiments show evidence of diffusive re-equilibration in response to Fe loss (higher $X_{Fo}$ contents than in the products of synthesis or shorter duration experiments) and diffusive homogenisation of melt FeO* contents ($X_{Fo}$ variance decreases from $2.9 \times 10^{-5}$ after 1 h to $9.6 \times 10^{-6}$ after 96 h). Indeed, olivine crystals in the products of the 24-h experiment appear to show diffusion profiles that are absent from the products of shorter experiments (Supplementary Fig. 1).

Plagioclase crystals occur in the products of the synthesis experiment on the Háleyjabunga analogue but not the Stapafell analogue (Figs. 2 and 8b). Plagioclase is also only stable within the portions of magma–magma reaction experiments initially derived from the Háleyjabunga lava analogue. In line with published phase equilibria[7], plagioclase crystals have uniformly high anorthite contents [$X_{An}$ ~ 0.83, where $X_{An} = Ca/(Ca + Na + K)$ on a molar basis]. Although $X_{An}$ variability exceeds analytical

uncertainty in the products of all magma–magma reaction experiments (Fig. 8b), likely because of rapid crystal growth that is also reflected in skeletal textures (Figs. 2 and 4)[54], mean $X_{An}$ does not vary systematically as a function of either position or experimental duration. Plagioclase stability, on the other hand, depends strongly on experimental duration. While plagioclase is stable throughout the portion of the 1-h experiment derived from the synthesis experiment on the Háleyjabunga lava analogue (Fig. 3a), regions where plagioclase is stable progressively shrink as experimental duration increases (Fig. 3b–d), leaving >1-mm broad regions of plagioclase resorption in the products of 24- and 96-h experiments. Projecting regions of plagioclase stability onto time-normalised composition profiles indicates that plagioclase resorption occurs when melt $Al_2O_3$ contents decrease below ~16 wt.% by diffusive re-equilibration within the melt (Fig. 6a). The region of plagioclase stability in the products of 24-h experiments is smaller than expected from comparisons with the products of other experiments, which we attribute to the modification of melt compositions by Fe loss to capsule materials leading to corresponding increases in other components including $Al_2O_3$. Nevertheless, our experiments indicate that plagioclase stability is dominantly controlled by melt $Al_2O_3$ content rather than melt CaO and $Na_2O$ contents (or melt Ca/Na) as both CaO and $Na_2O$ diffuse considerably faster than $Al_2O_3$ (Fig. 7)[48].

Clinopyroxene crystals occur in the products of synthesis experiments on both lava analogues (Fig. 2). While much variability in clinopyroxene Mg-number [$Mg\#_{cpx}$, where $Mg\#_{cpx} = Mg/(Mg + Fe)$ on a molar basis] is related to sector zoning[55], a slight difference in mean $Mg\#_{cpx}$ between the products of the experiments on the Háleyjabunga and Stapafell analogues (0.835 versus 0.850, respectively) reflects differences in their melt FeO* contents (Fig. 8c). However, differences between the two magma analogues are more clearly expressed in terms of

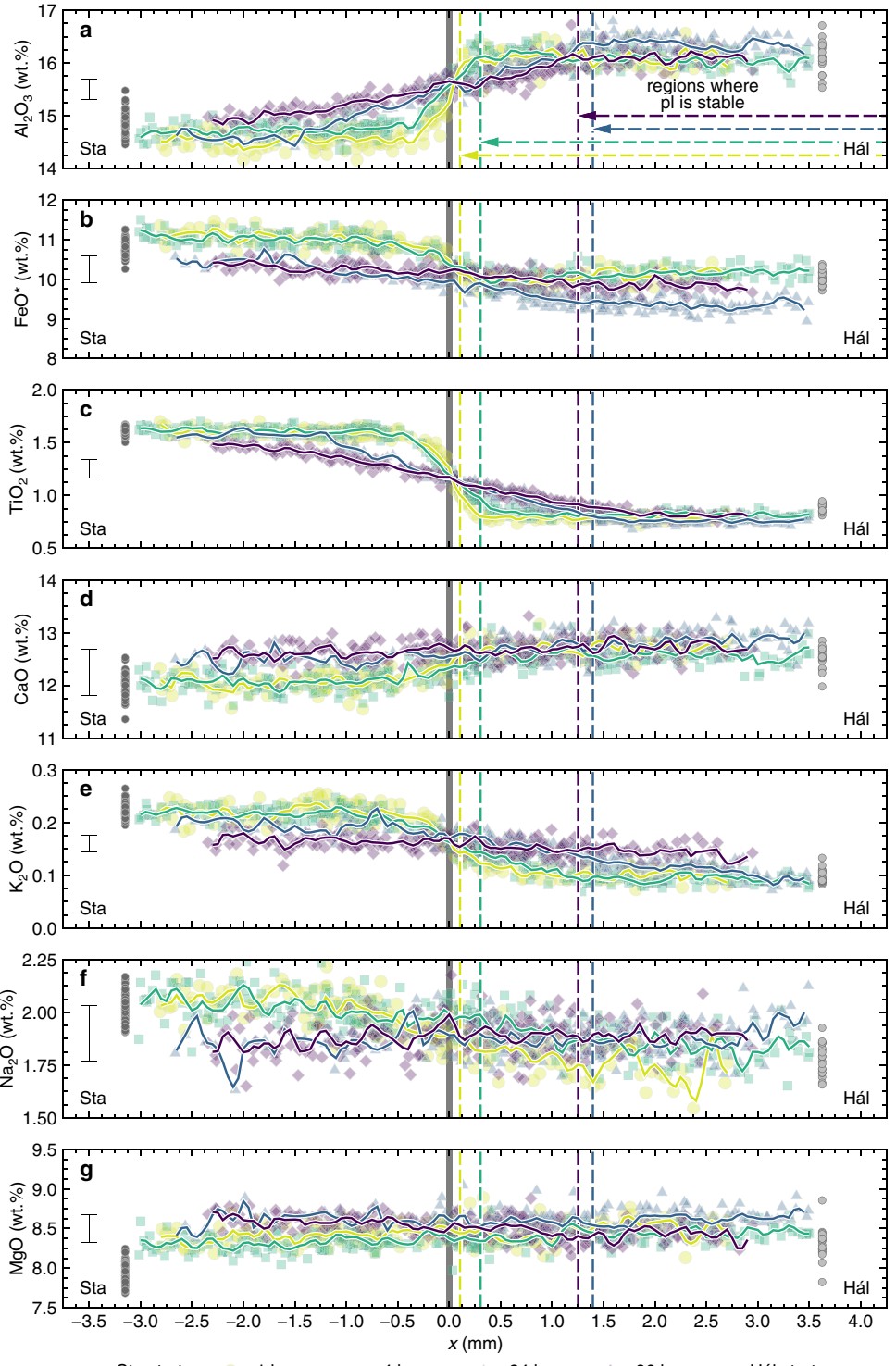

mean clinopyroxene $TiO_2$ contents, with clinopyroxene crystals grown from the incompatible element-enriched Stapafell analogue containing considerably more $TiO_2$ than those grown from the incompatible element-depleted Háleyjabunga analogue (~0.48 versus ~0.28 wt.%, respectively; Fig. 8d). Chemical variability associated with clinopyroxene sector zoning masks $Mg\#_{cpx}$ steps across magma–magma interfaces in the products of magma–magma reaction experiments (steps of ~0.01 are difficult to resolve within total $Mg\#_{cpx}$ ranges of 0.825–0.855, Fig. 8c). Nevertheless, overall gradients of low $Mg\#_{cpx}$ in the Stapafell analogue increasing to high $Mg\#_{cpx}$ in the Háleyjabunga analogue

can still be discerned. Steps in clinopyroxene $TiO_2$ contents between magma analogues are much more distinct (from ~0.2–0.3 wt.% in portions derived from the Háleyjabunga analogue to ~0.4–0.6 wt.% in portions derived from the Stapafell analogue; Fig. 8d), consistent with the slow intra-crystalline diffusion of $TiO_2$ during our experiments[56]. Positions of $TiO_2$ steps do, however, vary with experimental duration, with high-$TiO_2$ clinopyroxene crystals seemingly replacing low-$TiO_2$ clinopyroxene crystals in portions of the Háleyjabunga analogue adjacent to magma–magma interfaces in the products of longer-duration experiments.

**Fig. 5 Glass composition profiles through the products of magma–magma reaction experiments.** Glass composition profiles are centred on original magma–magma interfaces, which are indicated by vertical grey lines. Analyses from the products of experiments with different durations are shown with different colours and symbols. Solid lines show moving averages calculated by applying a Gaussian filter with a 0.25-mm bandwidth to raw analyses. Regions of the experimental products where plagioclase (pl) is stable are shown with dashed vertical lines and horizontal arrows. Compositions of glasses in the products of synthesis experiments are shown with grey symbols. Hál and Sta refer to Háleyjabunga and Stapafell lava analogues, respectively. Characteristic 2σ analytical uncertainties are shown. **a–f** Glass composition profiles in approximate order of increasing element diffusivity[48]: $Al_2O_3$ (**a**), $FeO^*$ (**b**), $TiO_2$ (**c**), $CaO$ (**d**), $K_2O$ (**e**) and $Na_2O$ (**f**). **a–c** Relatively slow-diffusing elements typically show composition profiles with sigmoidal shapes for all experimental durations. Sigmoidal profiles are especially clear in the cases of $Al_2O_3$ (**a**) and $TiO_2$ (**c**) but obfuscated by Fe loss for experimental durations ≥24 h in the case of $FeO^*$ (**b**). **d–f** Relatively fast-diffusing elements only show composition profiles with sigmoidal shapes for short experimental durations. This is emphasised most by $Na_2O$ (**f**) for which initial variability is erased within 24 h. **g** MgO shows little variability along compositional profiles beyond analytical uncertainty; small offsets between the products of synthesis and reaction experiments indicate that the former experienced slightly lower temperatures than the latter.

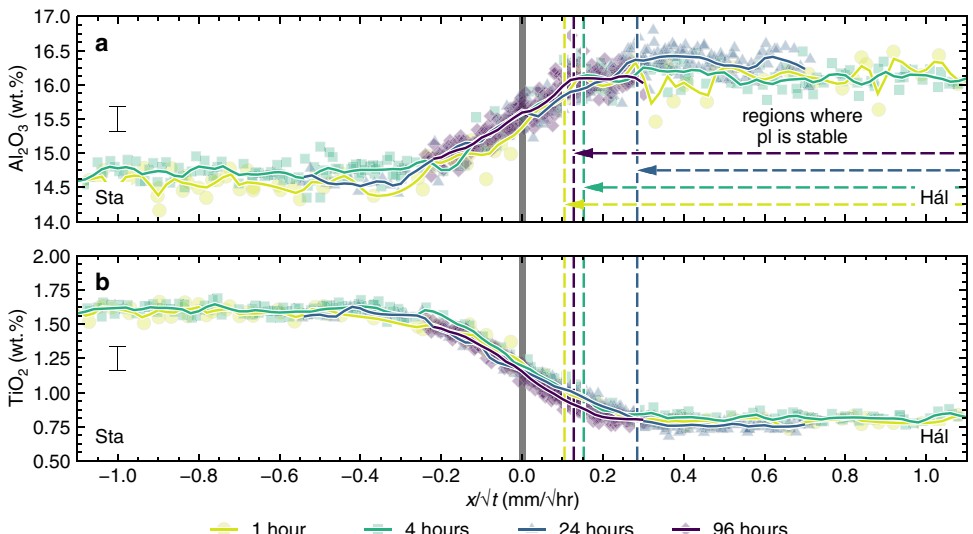

**Fig. 6 Time-normalised glass composition profiles through the products of magma–magma reaction experiments.** Normalising glass composition profiles by the square root of experimental duration eliminates the effects of time on the evolution of diffusively controlled profiles, which thus collapse onto single curves for each element[34,48]. Symbols are the same as in Fig. 5. Hál and Sta refer to Háleyjabunga and Stapafell lava analogues, respectively. **a, b** Closely overlapping time-normalised glass compositions profiles for $Al_2O_3$ (**a**) and $TiO_2$ (**b**) confirm that chemical variability in the products of magma–magma reaction experiments was dominantly controlled by diffusion within experimental melts. Equivalent profiles are shown for all elements in Supplementary Fig. 3.

## Discussion

**Diffusive controls over melt compositions.** Although physical mingling is vital for homogenising mantle-derived chemical variability in primitive basalts, diffusion is ultimately required for all phases to attain equilibrium. In line with dynamic mixing experiments[30–32], our magma–magma reaction experiments suggest that chemically variable magmas must be mechanically thinned to filaments no more than a few mm wide for diffusive homogenisation to be achieved within the day-long timescales associated with mixing processes in basaltic plumbing systems[24,25]. Indeed, chemical variability in olivine-hosted melt inclusions from individual eruptions records the entrapment of variably mixed melts, suggesting that crystallisation and diffusive homogenisation occur over broadly similar timescales[8,15]. Our experiments also demonstrate that mixing can fractionate elements with different diffusivities (e.g., $K_2O$ and $TiO_2$), potentially complicating records of mantle-derived chemical variability by processes described extensively in alkalic and evolved systems if not basaltic ones[30,34,57,58]. Moreover, diffusive fractionation has been implicated in creating high-field strength elements depletions in plagioclase-hosted melt inclusions from oceanic basalts[19,20], and it is at the μm-scale of melt inclusions that diffusion will have its greatest effects. Thus, even transient modifications of melt compositions by diffusion could fundamentally skew our understanding of magmatic processes if they are captured by the melt inclusion archives upon which geological interpretations are often based[59].

Our estimated effective binary diffusion coefficients are not significantly affected by the presence of crystals at mass fractions of ~0.1–0.3. Error function fits through glass composition profiles reproduce compositions from crystal-rich portions of experimental products equally well as those from crystal-poor portions (Supplementary Fig. 2), and diffusivities estimated from these fits are consistent with those estimated from melt-only systems (Fig. 7)[35–38,46,47]. Furthermore, melt compositions close to initial magma–magma interfaces are not detectably affected by either plagioclase resorption or clinopyroxene re-crystallisation but are dominantly controlled by diffusion within the melt, with Fe loss to capsule materials imposing second-order controls in longer-duration experiments. Our experiments thus suggest that the timescales over which basaltic magmas diffusively homogenise can be approximated by the re-equilibration of pure melts.

**Mineral responses to diffusive changes in melt compositions.** Primitive oceanic basalts inherit considerable chemical variability from the mantle[3–5], meaning that different batches of mantle-derived magma often have different bulk compositions at similar temperatures and degrees of magmatic evolution[8,9]. As a

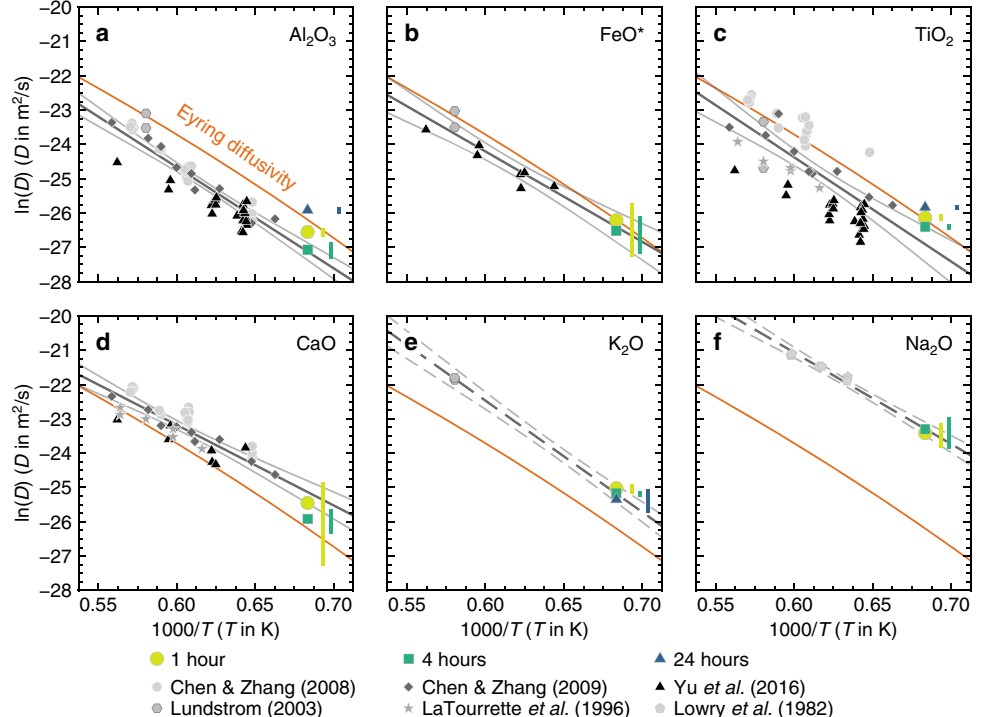

**Fig. 7 Element diffusivities in basaltic melts.** Effective binary diffusion coefficients ($D$) were estimated by fitting error functions to glass composition profiles to solve Fick's $2^{nd}$ Law, and are summarised in $1000/T$ versus $\ln(D)$ space alongside published diffusivities from experiments on basaltic systems[35–38,46,47]. Diffusion coefficients estimated from our experiments are only presented for element-duration combinations for which composition profiles were clearly sigmoidal in form. Temperature- and viscosity-dependent Eyring diffusivities calculated with a characteristic diffusive jump length of 0.4 nm are shown as orange lines[34,51]. Grey lines show regressions with 95% confidence intervals through global datasets. Dashed lines show tentative regressions for elements with sparse diffusion data. Vertical coloured bars show $1\sigma$ uncertainties in diffusion coefficients that are offset for clarity. **a** Estimated $Al_2O_3$ diffusivities are broadly consistent with Eyring diffusivities and slightly faster than from published diffusivities[36–38,46]. **b** Estimated $FeO^*$ diffusivities are similar to Eyring diffusivities and consistent with published diffusivities[38,46] **c** Estimated $TiO_2$ diffusivities are similar to Eyring diffusivities and consistent with many published diffusivities, though published values span up to three ln units at any given temerature[35–38,46]. **d** Estimated $CaO$ diffusivities are slightly faster than Eyring diffusivities but appear slightly slower than published diffusivities[35–38]. **e** Estimated $K_2O$ diffusivities are a ln unit faster than Eyring diffusivities but define a plausible array in $1000/T$ versus $\ln(D)$ space with the limited number of published diffusivities available[46]. **f** Estimated $Na_2O$ diffusivities are three ln units faster than Eyring diffusivities. They are also coherent with some published diffusivities[47], but not others[37].

consequence, isothermal mixing between chemically variable magmas, as recorded by melt inclusion suites[8,15], is just as likely to occur as mixing between cool resident magmas and hot recharge magmas[17,18,24,27]. Thus, crystal cargoes will not only experience recharge-induced changes in temperature but also mixing-induced changes in carrier liquid chemistry under near-isothermal conditions. In turn, these changes in chemistry create disequilibrium conditions that drive isothermal crystal resorption and growth by, for example, constitutional undercooling[39,40].

Plagioclase crystals in our magma–magma reaction experiments responded to diffusion-induced changes in local melt compositions by resorbing completely within hours. While plagioclase crystals are typically euhedral to subhedral, and often skeletal in experimental products derived from the Háleyjabunga analogue that are unaffected by diffusion within the melt (Figs. 2a and 4a, b), those at the limits of plagioclase stability in the products of magma–magma reaction experiments are typically anhedral and cuspate (Fig. 9). Our magma–magma reaction experiments have thus captured plagioclase resorption in action. Specifically, plagioclase mass fractions decrease from far-field values of 0.11 to 0 over distances of no more than ~200 μm. These short distances imply that plagioclase stability in our magma–magma reaction experiments was controlled by the kinetics of $Al_2O_3$ diffusion within the melt rather than plagioclase

resorption at crystal-melt interfaces. In other words, the distribution of plagioclase crystals in our experimental products indicates that once local melt $Al_2O_3$ contents dropped below ~16 wt.%, plagioclase crystals up to 20 μm in length resorbed completely within a few hours. Encouragingly, this observation is consistent with published experiments demonstrating that anorthite in basaltic liquids can resorb at >10 μm/h[38].

Olivine and clinopyroxene crystals also responded to diffusion-induced changes to local melt compositions, but in different ways from plagioclase crystals. The development of compositional zoning in olivine crystals from longer-duration experiments suggests that they at least partly responded to isothermal changes in melt composition by diffusive re-equilibration in the solid state (Fig. 8a and Supplementary Fig. 1). Indeed, the partial re-equilibration of olivine crystals over lengthscales of ~10 μm within 24 h is broadly consistent with a characteristic diffusion lengthscale (i.e. $x$, where $x \propto \sqrt{Dt}$)[45] of ~5 μm estimated using an Mg–Fe interdiffusion coefficient after Dohmen and Chakraborty[60], and provides independent validation of mixing timescales estimated from diffusion profiles in natural systems[25]. In contrast, clinopyroxene crystals seemingly responded to changes in local melt composition by resorption and re-crystallisation, as evidenced by steps in clinopyroxene $TiO_2$ content that progressively invade portions of experimental products derived from the Háleyjabunga analogue with increasing experimental

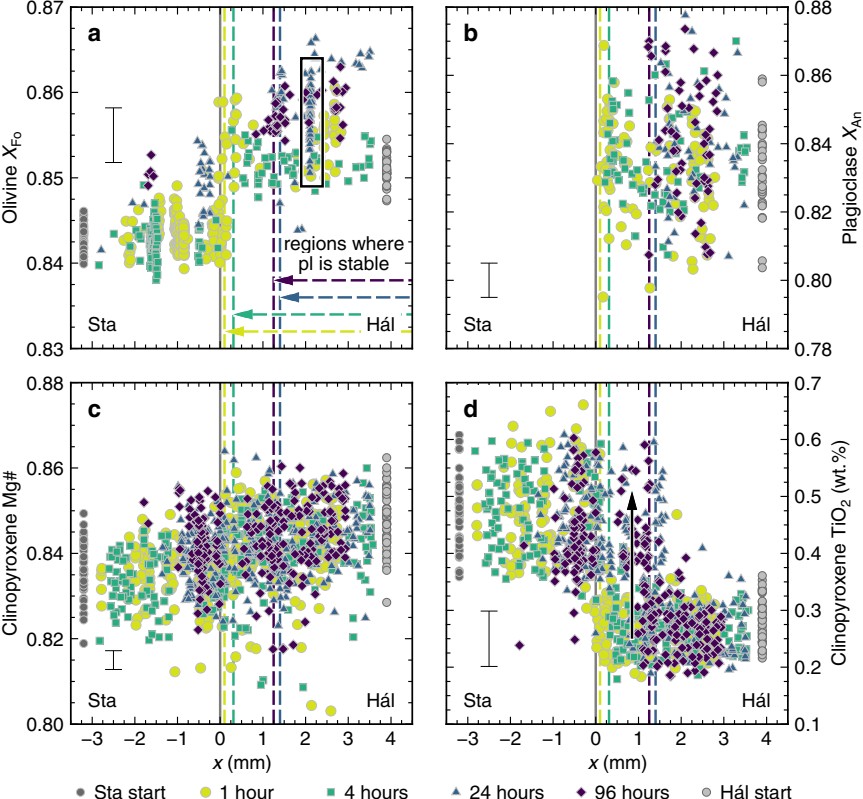

**Fig. 8 Mineral compositions in the products of magma–magma reaction experiments.** The positions of interfaces between juxtaposed magma cylinders are shown as vertical grey lines. Regions of the experimental products within which plagioclase (pl) is stable are shown with dashed vertical lines and horizontal arrows. Mineral compositions in the products of synthesis experiments are shown with grey symbols. Hál and Sta refer to Háleyjabunga and Stapafell lava analogues, respectively. Characteristic $2\sigma$ analytical uncertainties are shown. **a** Distribution of olivine compositions across the products of magma–magma reaction experiments summarised in terms of olivine forsterite content [$X_{Fo}$, where $X_{Fo} = Mg/(Mg + Fe)$ on molar a basis]. Steps in $X_{Fo}$ across interfaces between magma analogues reflect the higher FeO* content of the incompatible element-enriched Stapafell lava analogue with respect to the incompatible element-depleted Háleyjabunga analogue. Iron loss and diffusive re-equilibration of melt FeO* contents appear to have triggered the diffusive re-equilibration of $X_{Fo}$ in some crystals (one example is outlined with a black box; Supplementary Fig. 1). **b** Distribution of plagioclase compositions across the products of magma–magma reaction experiments summarised in terms of plagioclase anorthite content [$X_{An}$, where $X_{An} = Ca/(Ca + Na + K)$ on a molar basis]. The region where plagioclase is stabile contracts significantly as a function of increasing experimental duration as a result of plagioclase resorption in response to diffusively driven changes in melt compositions. Although $X_{An}$ is somewhat variable, potentially as a consequence of rapid crystal growth during synthesis experiments[54], it does not vary systematically with position or experimental duration. **c** Distribution of clinopyroxene compositions across the products of magma–magma reaction experiments summarised in terms of clinopyroxene Mg-number [$Mg\#_{cpx}$, where $Mg\#_{cpx} = Mg/(Mg + Fe)$ on a molar basis]. While much variability in $Mg\#_{cpx}$ reflects the development of clinopyroxene sector zoning[55], mean $Mg\#_{cpx}$ contents are slightly higher in the FeO*-poor Háleyjabunga analogue than the FeO*-rich Stapafell analogue. **d** Distribution of clinopyroxene compositions across the products of magma–magma reaction experiments summarised in terms of clinopyroxene $TiO_2$ contents. Clinopyroxene crystals from the incompatible element-enriched Stapafell analogue are richer in $TiO_2$ than those from the incompatible element-depleted Háleyjabunga analogue. Moreover, clinopyroxene crystals within regions of plagioclase resorption have high $TiO_2$ contents, suggesting that plagioclase resorption is spatially correlated with the resorption and re-crystallisation of clinopyroxene, as indicated by the black arrow.

duration (Fig. 8d). Importantly, these steps in clinopyroxene $TiO_2$ content are offset from magma–magma interfaces and coincide with diffusively controlled plagioclase resorption fronts, suggesting that they are not advective in origin. Moreover, clinopyroxene is denser than the melts we investigated (~3.3 versus ~2.7 g/cm³), meaning that high-$TiO_2$ clinopyroxene could not have floated from the Stapafell analogue into the overlying Háleyjabunga analogue. The implications of our experimentally observed clinopyroxene re-crystallisation for natural systems remain unclear, however, partly because it is difficult to gauge how large natural crystals (>100 μm long) would respond from observations on small experimentally produced crystals (≤10 μm long). It is nevertheless plausible that well-documented instances of isobaric clinopyroxene resorption in natural basalts record isobaric changes in magma chemistry as much as changes in magma temperature[12,61].

**Magma mixing creates and modifies basaltic crystal cargoes.**
Mush disaggregation creates the crystal cargoes carried by many oceanic basalts[15,62,63], and the eruption of plagioclase glomerocrysts with quasi-cumulate textures is often interpreted as evidence of pre-eruptive crystal entrainment[21–23]. Although numerical modelling provides vital insights into the dynamics of crystal-rich magmas[64], the processes by which initially cohesive mushes disaggregate remain largely elusive. For example, primitive crystal cargoes are generally too refractory to be chemically excavated from cohesive mushes by the relatively evolved liquids that often carry them to the surface[15,22,62]. That is, assuming evolution along a single liquid line of descent, the injection of liquids in equilibrium with relatively evolved plagioclase crystals ($X_{An} \leq 0.7$) would not unlock mushes of primitive plagioclase crystals ($X_{An} > 0.8$) but rather lock them up further by triggering the crystallisation of overgrowth rims. In nature, however,

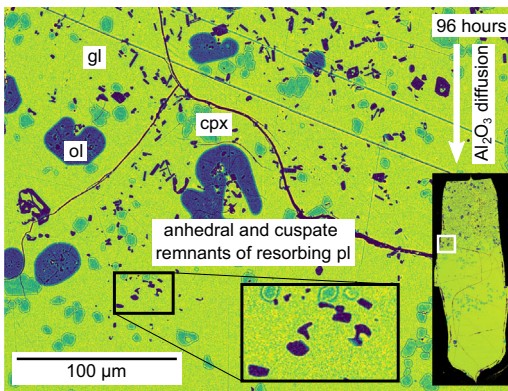

**Fig. 9 False-colour backscattered electron (BSE) image showing plagioclase resorption in the products of the 96-h magma–magma reaction experiment.** Many plagioclase (pl, purple) crystals close to plagioclase resorption fronts show resorbed textures, and the last remnants of resorbing plagioclase crystals are especially anhedral and cuspate (see inset). Glass (gl) is shown in yellow to green colours. Some olivine (ol, blue) crystals close to plagioclase resorption fronts are anhedral and embayed, whereas clinopyroxene (cpx, blue-green) crystals retain euhedral to subhedral textures.

erupted crystal cargoes and carrier liquids are often derived from chemically distinct primary melt distributions that have evolved along different liquid lines of descent[22,65,66]. It is thus feasible that mush-derived crystals will have never been in equilibrium with melts parental to their eventual carrier liquids. A key consequence of this is that mush disaggregation may be triggered by differences in magma composition as well as by differences in magma temperature.

Chemical and isotopic disequilibria between basaltic melts and mush-derived high-$X_{An}$ plagioclase crystals are well documented in oceanic basalts from ocean islands and mid-ocean ridges of all spreading rates[22,23,62,65,67,68], demonstrating that mixing between chemically variable primitive magmas is widespread in the oceanic realm[8,9]. Although pervasive evidence of mixing and disequilibrium does not in itself constrain the mechanisms by which mush disaggregation and crystal entrainment occur, it does highlight that the deep chemical variability required to trigger mush disaggregation by mixing-induced chemical disequilibrium and crystal resorption is globally present. Moreover, we speculate that variations in the $H_2O$ content of arc magmas may have analogous effects to variations in the major element content of oceanic basalts, whereby the injection of $H_2O$-rich magmas may facilitate mush disaggregation by triggering plagioclase resorption[69].

Very few basalts erupted in oceanic settings are in equilibrium with high-$X_{An}$ plagioclase at any stage of their evolution[15,62,70]. Instead, high-$X_{An}$ plagioclase in oceanic settings probably crystallises from incompatible element-depleted primitive melts that are correspondingly enriched in refractory elements like CaO and $Al_2O_3$, and rarely survive crustal processing to erupt at the surface[7,70]. Indeed, these melts probably stall at depth where they form plagioclase-rich mushes and cumulates. Our magma–magma reaction experiments on chemically variable but naturalistic analogues of erupted Icelandic lavas suggest that injecting incompatible element-enriched and plagioclase-undersaturated magmas (like the Stapafell lava analogue) into high-$X_{An}$ plagioclase-rich mushes (like the Háleyjabunga lava analogue but with a higher crystal fraction) could trigger mush disaggregation without requiring hot recharge. This is important because primitive crystal cargoes in oceanic basalts typically show evidence of being entrained by cool and evolved magmas rather

than hot and primitive ones[22,23,62]. We thus propose that the replacement of plagioclase-saturated mush liquids with plagioclase-undersaturated liquids can drive resorption within interstices and along grain boundaries that in turn unlocks mushes and facilitates crystal entrainment by ascending magmas (Fig. 10a).

While advection almost certainly drives magma mixing and mush melt replacement on the macro scale (>1 mm), diffusion is vital for changing melt compositions on the micro scale (<1 mm) of crystal-melt interactions that are ultimately responsible for determining crystal textures and controlling mush cohesion[31,32]. Although the lengthscales of plagioclase resorption we observe are short (~10 μm), partly reflecting the small size of plagioclase crystals in our experimental products with respect to natural samples (~10–20 versus >100 μm long), even modest amounts of resorption within interstices and along grain boundaries could trigger disaggregation[41]; large crystals do not need to resorb completely for mush cohesion to be lost, they merely need to become sufficiently detached to be entrained by their surrounding melts. That only modest amounts of resorption could enable mush disaggregation also bears on the relative volumes of different melts required to facilitate entrainment. Our lava analogues were reacted in approximately 50:50 proportions, meaning that melt $Al_2O_3$ contents would eventually reach sufficiently low levels throughout the experimental samples for all plagioclase crystals to resorb (i.e. below ~16.0 wt.%). Conversely, if the proportion of incompatible element-enriched lava analogue were below ~0.25 then plagioclase would be stable upon complete re-equilibration (i.e. melt $Al_2O_3$ contents would exceed ~16 wt.%), re-crystallising in regions where it had previously resorbed, albeit with a lower $X_{An}$. However, even transient resorption triggered by volumetrically minor injections of incompatible element-enriched magmas could permit the one-way process of disaggregation, with subsequent re-crystallisation feasibly contributing to the textural complexity of erupted crystals[21,23].

Regardless of their origins, crystal cargoes will be modified as physical mixing processes transfer them between liquids with different phase equilibria. The transfer of high-$X_{An}$ plagioclase crystals between melts with different CaO and $Al_2O_3$ contents can thus result in either growth or resorption depending on the direction of transfer. For example, our magma–magma reaction experiments at 1190 °C demonstrate how the diffusive equilibration of incompatible element-depleted and plagioclase-saturated magmas with incompatible element-enriched and plagioclase-undersaturated magmas can trigger isothermal plagioclase resorption in the former without plagioclase nucleation in the latter (Figs. 4 and 8). At lower T conditions (~1170 °C[7]), however, isothermal $Al_2O_3$ diffusion may trigger plagioclase nucleation in initially plagioclase-free and incompatible element-enriched magmas rather than plagioclase resorption in initially plagioclase-bearing and incompatible element-depleted magmas because the former would be closer to plagioclase saturation than is the case for the experiments described here. In plagioclase-saturated but chemically variable systems at yet lower T conditions (≤1160 °C[7]), isothermal $Al_2O_3$, CaO and $Na_2O$ diffusion could produce plagioclase crystals with $X_{An}$ zonation. Comparable differences in behaviour could also arise by mixing magmas initially equilibrated at different pressures because plagioclase stability correlates with P[71].

Although decompression, primitive recharge and boundary layer effects doubtlessly contribute towards the textural complexity of plagioclase crystals carried by oceanic basalts[15,18,21,23], isothermal and isobaric changes in melt composition are also likely to be important. Indeed, mixing-induced changes in melt composition may drive the cycles of resorption and

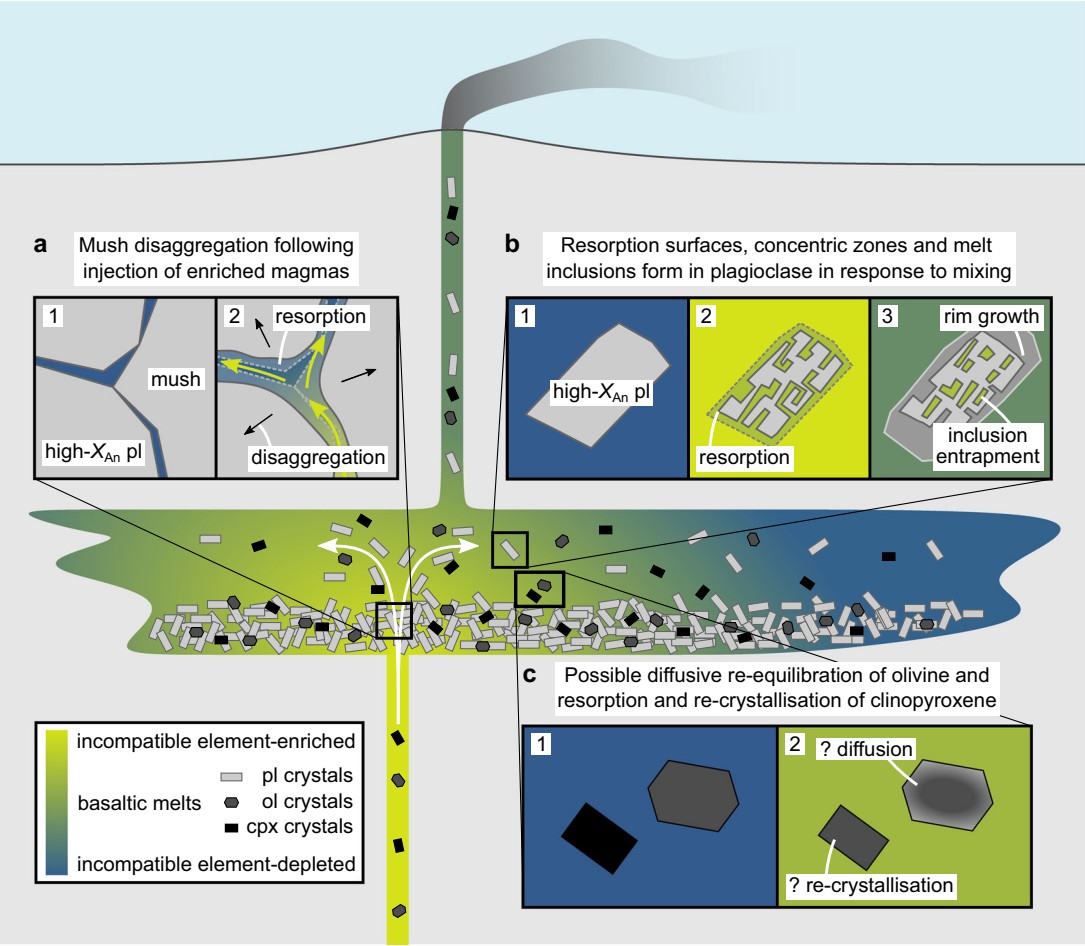

**Fig. 10 Cartoon summarising how mixing-induced chemical disequilibrium creates and modifies basaltic crystal cargoes.** Primitive and incompatible element-depleted magmas crystallise high-anorthite [high-$X_{An}$, where $X_{An} = Ca/(Ca + Na + K)$ on a molar basis] plagioclase (pl) alongside olivine (ol) and clinopyroxene (cpx) during storage in the crust[7,70]. Even under isothermal conditions, recharge by incompatible element-enriched and plagioclase-undersaturated primitive magmas can trigger changes in the crystalline portions of incompatible element-depleted and plagioclase-saturated magma reservoirs. **a** The infiltration of plagioclase-dominated mushes by plagioclase-undersaturated liquids may trigger mush disaggregation by driving resorption within interstices and along grain boundaries, as indicated by chemical and isotopic disequilibria between some basaltic melts and their cargoes of high-$X_{An}$ plagioclase crystals[22,65]. **b** The transfer of high-$X_{An}$ plagioclase crystals between variably incompatible element-enriched magmas that are also variably saturated in plagioclase produces resorption surfaces, concentric zoning, overgrowth rims and melt inclusions[19,20,23]. **c** Olivine and clinopyroxene crystals may re-equilibrate, and resorb and re-crystallise, respectively, in response to isothermal mixing of variably incompatible element-enriched magmas with different major element compositions.

re-crystallisation implicated in the formation of plagioclase-hosted melt inclusions (Fig. 10b)[19,20]. Although correlated changes in temperature and melt composition will drive the largest diffusive changes in olivine $X_{Fo}$ contents[24–26], $X_{Fo}$ variations may also reflect isothermal variations in melt chemistry (Figs. 8a and 10c). Indeed, our experiments suggest that olivine crystals respond to changes in melt composition by diffusive re-equilibration rather than re-crystallisation. Conversely, our observations suggest that clinopyroxene crystals can respond to changes in melt composition, at least over short lengthscales, by resorption and re-crystallisation rather than diffusive re-equilibration (Fig. 10c). Some of the textural complexity observed in natural clinopyroxenes may thus result from mixing between chemically distinct primitive basalts[12,61].

Magma mixing creates and modifies basaltic crystal cargoes and can therefore alter magmatic plumbing system dynamics and erupted records of mantle-derived chemical variability. We therefore argue that crystal resorption in response to mixing-induced chemical disequilibrium represents a currently under-appreciated yet potentially widespread mechanism for

disaggregating crystal mushes that can also alter the viscosity, density and eruptibility of oceanic basalts through the entrainment of buoyant high-$X_{An}$ plagioclase crystals in large volumes (sometimes > 30 vol.%)[22,62]. Overall, isothermal mixing between chemically variable primitive magmas is likely to play an equally important role in generating the texturally and chemically complex crystal cargoes we observe at the surface as the more widely recognised process of primitive recharge.

## Methods

**Experimental methods.** Synthesis experiments were performed in an internally heated pressure vessel (IHPV) at the Institut für Mineralogie of the Leibniz Universität Hannover, Germany. Experiments were performed at 300 MPa and 1190 °C to maximise differences in resulting phase assemblages and proportions according to published equilibrium phase relations[7]. Synthetic analogues of the incompatible-element depleted Háleyjabunga and incompatible element-enriched Stapafell lavas were prepared by Neave et al.[7] from reagent grade oxide and carbonate powders that were fused twice in Pt crucibles placed in a muffle furnace at 1600 °C. Each fusion was performed for 1 h, after which the melt was quenched by pouring it onto a clean brass plate. Quenched glass chips were then powdered in an agate disc mill to ensure that starting materials were compositionally homogenous. About 300 mg of each powdered starting material was then loaded into $Au_{80}Pd_{20}$

capsules with external and internal diameters of 2.6 and 2.2 mm, respectively, and welded shut. Each capsule was ~25 mm long in order to maximise the volumes of synthetic magmas that could be produced in each IHPV experiment. Prior to being loaded with starting materials, capsules were pre-saturated with ~0.25 wt.% Fe to minimise Fe exchange between capsule materials and experimental samples[72]. This was done by electroplating the capsules and then annealing them for 48 h at 950 °C in a reducing $H_2$-Ar atmosphere following the procedures described by Husen et al.[71].

Prepared capsules were suspended from a Pt wire in the hot zone of the IHPV[73]. The IHPV was then pressurised to 300 MPa with Ar and heated to 1190 °C. Pressure was monitored with a strain gauge manometer and did not vary by more than 5 MPa; temperature was monitored with four unsheathed S-type thermocouples and did not vary by more than 5 °C. Experimental temperatures were approached by heating the furnace from room temperature to 10 °C below the target temperature at a rate of 50 °C/min; final heating from 1180 to 1190 °C was performed at a rate of 10 °C/min to avoid overshooting. For the magma synthesis experiments, thermal cycling (1190 ± 5 °C) was applied for the first 24 h to promote the growth of large crystals[74]. Temperature was then kept constant for the last 48 h. Experimental products were quenched after 72 h by fusing the Pt wire on which the capsules were suspended. The capsules dropped into a cold zone at the bottom of the vessel cooled at a rate of ~150 °C/s, which was sufficient to avoid the formation of quench crystals.

All experiments were conducted under nominally dry conditions (no $H_2O$ was added to the dried starting powders), which resulted in melt $H_2O$ contents of ~0.7 wt.% following the reduction of $Fe_2O_3$ in the starting glasses to FeO and the inward diffusion of trace $H_2$ from the Ar pressure medium at high temperatures. These $H_2O$ contents are consistent with the experiments having been run under broadly reducing conditions about one log unit above the fayalite-magnetite-quartz redox buffer (FMQ + 1): melt $H_2O$ contents of ~0.7 wt.% correspond to an $a_{H_2O}$ of ~0.06[75], which is related to the $f_{O_2}^{sample}$ by the relationship $f_{O_2}^{sample} = f_{O_2}^{vessel} \cdot a_{H_2O}^2$, where $f_{O_2}^{vessel}$ was estimated as FMQ + 3.3 based on the $f_{O_2}$ of samples saturated in a pure $H_2O$ fluid[76].

Magma–magma reaction experiments were performed under exactly the same conditions as synthesis experiments, and in the same IHPV. The products of each synthesis experiment were cut into four cylinders ~3.5 mm long, some of which were irregular because of the crimps used to seal capsule ends. The dimensions of these irregularities exceeded the dimensions of most diffusive features investigated here. The ends of the products of each synthesis experiment were retained to determine the phase assemblages and phase compositions present at the start of magma–magma reaction experiments. Magma cylinders from each synthesis experiment were then juxtaposed within new $Au_{20}Pd_{20}$ capsules with the same dimensions and approximate Fe contents as those used for synthesis experiments. Each capsule was then welded shut and magma–magma reaction experiments were performed in the same way as described for synthesis experiments. Capsules were oriented with cylinders synthesised from the relatively dense and FeO*-rich Stapafell lava analogue beneath those synthesised from the relatively light and FeO*-poor Háleyjabunga lava analogue to prevent intra-capsule convection[77]. Magma–magma reaction experiments were performed for durations of 1, 4, 24 and 96 h in order to capture the time-dependent nature of kinetic processes. The products of magma–magma reaction experiments were cut longitudinally and mounted in resin alongside the reserved products of synthesis experiments for subsequent imaging and analysis.

**Analytical methods**. Experimental products were imaged by field emission gun scanning electron microscopy (FEG-SEM) on a JEOL JSM-7610F instrument at the Institut für Mineralogie of the Leibniz Universität Hannover, Germany. Backscatter electron images (BSE) were typically collected using an accelerating voltage of 15 kV and a working distance of 15 mm. BSE maps of experimental products were acquired using Bruker's ESPRIT software.

The major and minor element compositions of experimental products were determined by electron probe microanalysis (EPMA) on a Cameca SX100 instrument at the Institut für Mineralogie of the Leibniz Universität Hannover, Germany. Silicon, Ti, Al, Cr, Fe, Mn, Mg, Ca, Na, K and P were measured in glasses with a beam size of 12 μm, an accelerating voltage of 15 kV and a current of 10 nA. Silicon, Ti, Al, Cr, Fe, Mn, Mg, Ca, Na and K were measured in minerals with a beam size of 1 μm, an accelerating voltage of 15 kV and a current of 15 nA. Elements were counted on peak for 20 s, with the exceptions of Si and Na that were counted on peak for 10 s to minimise detector drift and Na migration, respectively. Background counting times were half on-peak counting times. The following standards were used for calibration: wollastonite (Si and Ca), $TiO_2$ (Ti), jadeite (Al), $Cr_2O_3$ (Cr), $Fe_2O_3$ (Fe), $Mn_3O_4$ (Mn), MgO (Mg), albite (Na), orthoclase (K) and apatite (P). The following secondary standards were regularly analysed to correct for inter-session drift and to monitor accuracy and precision: VG-2 basaltic glass (NMNH 111240-52; using the preferred MgO content), Kakanui augite (NMNH 122142; using preferred values), San Carlos olivine (NMNH 111312-44) and Lake County plagioclase (NMNH 115900)[78]. Accuracy and precision were typically better than 2% and 2%, and 10% and 10% for major (>1 wt.%) and minor

(<1 wt.%) elements, respectively. Typical analyses of standards are provided alongside analyses of experimental products in Supplementary Data 7.

Glass $H_2O$ contents were determined in the products of the synthesis experiments and 1- and 96-h magma–magma reaction experiments by Fourier-transform infrared (FTIR) spectroscopy with a Bruker IFS88 instrument at the Institut für Mineralogie of the Leibniz Universität Hannover, Germany, following the methods described by Husen et al.[71]. In short, $H_2O$ contents were determined in ~100-μm thick wafers using the peak attributed to the OH stretch vibration (3550 cm$^{-1}$) using a molar absorption coefficient of 68 L/mol cm. Estimated glass $H_2O$ contents are provided in Supplementary Data 7 with further information about experimental $a_{H_2O}$–$f_{O_2}$ conditions.

**Estimating effective binary diffusion coefficients**. Effective binary diffusion coefficients ($D$) were estimated by fitting error functions to glass composition profiles through the products of magma–magma reaction experiments to solve Fick's 2$^{nd}$ Law: $C(x,t) = C_1 + \frac{C_0 - C_1}{2}\left(1 - \text{erf}\left(\frac{x}{2\sqrt{Dt}}\right)\right)$, where $C(x,t)$ is the concentration in wt.% of the diffusing element $C$ at distance $x$ in m after time $t$ in s, and $C_0$ and $C_1$ are the initial concentrations of the diffusing element on either side of the couple[49]. Fitting was performed by minimising the $\chi^2$ misfit associated with the following function: $y_{est} = a\text{erf}(b(x + c)) + d$, where $y_{est}$ is the predicted concentration of a given oxide in wt.% and $a$, $b$, $c$ and $d$ are fitting parameters. The $\chi^2$ misfit was defined as follows: $\chi^2 = \sum_{x=0}^{n}\left(\frac{(y_{obs} - y_{est})^2}{2\sigma}\right)$, where $y_{obs}$ is the observed concentration of a given oxide in wt.% at a distance of $x$, $y_{est}$ is the predicted concentration of a given oxide in wt.% at a distance of $x$ and $\sigma$ is the uncertainty associated with analyses of a given oxide in wt.%. Minimisations were performed with the fminsearch() function implemented in the pracma package of R[79]. Element diffusivities in m$^2$/s were then calculated using the following relationship[45]: $D = \frac{((1/b)^2)}{4t}$, where $b$ is the fitting parameter described above. Diffusion coefficients were only estimated for element-duration combinations for which composition profiles were clearly sigmoidal in form and had well-defined plateaux at each end. Thus, no diffusion coefficients were estimated from the products of the 96-h experiment or for some elements (FeO*, CaO and $Na_2O$) in the products of 24-h experiment. Uncertainties in diffusion coefficients were estimated by fitting composition profiles that had been repeatedly resampled according to the analytical uncertainties associated with each element.

## Data availability
The SEM images and EPMA data generated in this study are provided in Supplementary Data 1–6 and 7, respectively.

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

## Acknowledgements
We thank Ulrich Kroll and Stefan Linsler for their help with maintaining IHPV equipment, and Renat Almeev and Chao Zhang for their help with EPMA. This work and D.A.N. were supported by the German Research Foundation (DFG; NE2097/1-1). D.A.N. was also supported Presidential Fellowship from the University of Manchester and a NERC Independent Research Fellowship (NE/T011106/1).

## Author contributions
D.A.N. designed the project and secured funding. P.B. performed the experiments and EPMA analyses under the supervision of D.A.N. and F.H.; H.B. performed the FTIR analyses. All authors contributed to data interpretation. D.A.N. wrote the manuscript with contributions from P.B., H.B. and F.H.

## Competing interests
The authors declare no competing interests.
