## [Peer Review File · Nature Communications]

Reviewers' Comments:

Reviewer #1:

Remarks to the Author:

NCOMMS-21-01451

Dear authors and editors,

Thank you for giving me the opportunity to evaluate such work. I enjoy the manuscript and paid a lot of attention to all aspects covering my research area. I hope my comments will be perceived as constructive as possible.

The manuscript deals about isothermal mixing of chemically different basalts, with an experimental approach, and finds applications in magma remobilisation (although that term was not used by authors) in oceanic domains.

The manuscript is well constructed, illustrated, and written. What a pleasure to evaluate a mature work! It respects NCOMMS policies in terms of length, illustrations. The availability of all data as supplementary materials is also very appreciated.

I structured my review with major/minor comments, and other minor comments directly in the pdf file. I think the manuscript is very close to be ready for publication; there is one weakness to me (major comment 4) that may drive my final decision, according to the authors' answer.

Major points

1/ Despite the focus on chemical mixing of this study, a textural description of mixing experiments is lacking (Related to comment L.126). Olivine concentration decreases with experiment duration, therefore, MgO & FeO concentrations in the melt increase, and relatively, Al₂O₃ concentration (directly related to Plagioclase crystallization) decreases... Please, add a (short?) textural description of the products (additional figure with zoom of end-members at 96h?) that will be connected to the discussion about the chemistry of the system.

Did you notice the shape of the interface? e.g. 4h experiment with a triangle/conic shape.

I am also thinking about providing, if possible, an image showing the initial texture of syntheses, even if it is very close to 1h magma-magma interaction experiment (as supplementary figure?). It is important to appreciate the mineral phase proportion through experiment duration. Since you have crystal and melt chemistry, you may check if your system remains chemically closed by mass balance, and perhaps you will get some clues about iron behaviour.

2/ I wish to see some clarifications about plagioclase behaviour. I have doubts about the resorption of the plagioclase shown in Fig. 8. First because some of its faces remains crystalline (it is subeuhedral, like other in top right corner of the same figure close to the white arrow), and also because of the presence of euhedral plagioclase crystal in the same experimental product, visible on the attached file equivalent to Fig 3d at various distances from the interface. Similar plagioclase crystals are present in the four presented experiments. Some of those euhedral crystals contain embayments or melt inclusions, but the latter could result from a relatively fast crystal growth (works of Kirkpatrick 1975, Donaldson et al. 1976, Faure et al. 2003-2005, Laumonier et al. 2019 CMP). Could you please discuss this point and strengthen your argumentation?

Related to this point (and L293-295), the size of plagioclase crystals in the initial syntheses may give valuable information (I would like to see images with lower magnification of the syntheses). Do you think Ostwald ripening can affect plagioclase population? (see for instance Cabane et al., 2005).

3/ To be completely convinced about the evolution of olivine phase, I need some extra arguments.

First, I questioned the solid-state diffusive re-equilibration: what are the argument to reach this conclusion? I would agree, for instance, if several chemical profiles in olivine of different experimental durations were presented that show time-dependent reaction (from the shape of the profile, as you

did for melt in Fig 4). You could estimate Fe/Mg diffusion in olivine and compare with literature (partly done, L 303). Although I have no quantitative argument, I suggest the profiles presented in supplementary Fig. 1 result from the melt composition that varies in the course of the crystallization during syntheses experiments. I have access to a small area of each synthese with Fig. 2 (that's why an image of syntheses at lower resolution would be helpful), but there is no clue of crystal zoning from SEM image, contrary to some olivines in the 24h experiment.

4/ Paragraph L333: Main cause of crystal mush disaggregation of oceanic basalts. This paragraph (+ consequences and the last paragraph) appears to me as the kernel of the study as it will drive my opinion about whether or not this contribution is relevant to a large audience reached by NCOMMS. In brief, the authors show very well the consequence of isothermal mixing from experiments. The last obstacle to me is to demonstrate if this process is an alternative to classical reheating of magma reservoir potentially leading to crystal mush disaggregation or if it the main process occurring in oceanic magma reservoirs. In the latter case, the message is original. However, the demonstration of the importance of such process is not strong enough to me. The demonstration holds to the sentence L341: "We thus suggest that chemical rather than thermal disequilibrium may play an important role in triggering crystal mush disaggregation in some magma reservoirs " and L396 sums up my feeling: "Overall, isothermal mixing between chemically variable primitive magmas likely plays an equally important role in generating the texturally and chemically complex crystal cargoes we observe at the surface as more widely recognised processes invoking mixing between variably evolved and thermally distinct magmas." Consequently, to me, the title of the study should mention that last point.

Minor points

1/ Fo content of olivine, L192 "Steps in X_{Fo} also occur in the products of 24- and 96-hour experiments". Actually, I do not agree. To me, there is insufficient data close to the interface to conclude. One could even draw a line from the data point at x=-3mm to the ones at x=3.5mm (24h experiments, Fig.7) that follows the olivine Fo distribution. Can you argument better to support your opinion, or be more careful if you agree with me?

2/ From L323 to the end of the paragraph. "liquids in equilibrium with relatively evolved plagioclase crystals ($X_{An} \leq 0.7$) could not resorb grain boundaries in mushes dominated by primitive plagioclase crystals ($X_{An} > 0.8$) formed during earlier phases of higher temperature crystallisation" (L325-328). Why not? this seems counterintuitive to me; the higher the chemical potential, the faster the chemical reaction. I would rather interpret glomerocrysts as colder and solidified (i.e. cohesive, resistant) enclaves such as the so common mafic enclaves found in evolved magmas (although the temperature difference between mafic magma and more felsic host may be higher than for the case studied here). Can you nuance your opinion or provide additional argumentation, reference to support it?

3/ L 352. I do not agree with this sentence and ask you to strengthen the argumentation. To me, crystal-melt disequilibria evidence magma mingling/mixing, but not that chemical variability plays a widespread role in driving crystal mush disaggregation. Same for L358: "Nevertheless, the occurrence..."

4/ As pointed out also in the abstract, please describe "oceanic basalts" in the discussion and the different areas of applicability of your study.

5/ What are the yellowish (saturated) "objects" in some figures? Oxides? Are they present in the syntheses? e.g. in the 24h experiment, close to the interface, there is an olivine with 2 melt inclusions and another yellow inclusion/area. Another one, with square shape in the 4h experiment, at ~1/3 from the bottom of the sample, left side, and more frequently in the same experiment, few 10 μm above the interface.

Best regards,
Mickael Laumonier
2021 February 3rd

Reviewer #2:

Remarks to the Author:

In their paper entitled "Mixing-induced chemical disequilibrium creates and modifies basaltic crystal cargoes", Neave et al. report new experimental observations of isothermal magma mixing-induced chemical disequilibrium in basaltic melts and how such mixing can create and modify crystal cargoes. The authors have performed magma-magma reaction experiments at 300 MPa and 1190 °C on synthetic analogues of two types of Icelandic lavas (with distinct major element compositions) for several durations (from 1 to 96 hours), and then performed careful analysis of the experimental products. They found that the melt (near interface) experienced diffusive re-equilibration with elemental diffusivities overlapping with those reported in the literature, and different minerals reacted in different ways to the isothermal mixing: most olivines survived through diffusive re-equilibration while clinopyroxene and plagioclase had partial and complete resorption respectively after a few hours. These results highlight the potentially significant yet overlooked effect of isothermal mixing-induced chemical disequilibrium on crystal mush disaggregation, chemical/physical properties of magmas, and the eruption dynamics, and thus have wide implications in the studies of mantle chemistry as well as magma mixing and pre-eruptive magmatic processes in basaltic systems.

Overall, I find the conclusions and data interpretation in this paper robust and convincing. The abstract is concise, and the main text is well written with clear structure. Experimental procedures and analytical methods are documented in detail, and figures are illustrated with high quality. Considering all these factors, I am glad to recommend this paper for publication in Nature Communications (with minor revisions). The only question/doubt I have in mind is whether the mineral responses observed in this study (especially recrystallisation/resorption of clinopyroxene/plagioclase) would be different under other P-T-melt/mineral-composition conditions? It would be useful to comment about this in discussion. The rest of my comments and some grammatical edits are listed below.

- Line 58-59: "...at a under isothermal contisions,..." should be "...under isothermal conditions,..."?
- Line 173-176: I am wondering the range of melt water contents that were used to determine diffusivity in previous studies, and whether this has a large influence on the difference between the diffusivities reported in the literature and your estimation.
- Line 180: not sure what this sentence means. Please correct it.
- Lines 182-183: better to report the value of viscosity used for calculation if it is not shown elsewhere in the manuscript.
- Lines 183-185: "...possibly reflecting differences in diffusion mechanisms between network-forming and network-modifying cations.": How about FeO, which can be also regarded as network-modifying component but shows diffusivities similar to the theoretical Eyring diffusivities in Fig. 6?
- Line 200 and also Fig. 9 caption: the equation for X_{An} is usually written as $X_{An} = Ca/(Ca+Na+K)$, unless it is defined differently in this study.
- Line 287: "...no more that..." should be "...no more than..."
- Line 324: "...too refractory to have been thermally by...": a verb is missing between "thermally" and "by".
- Line 341: "...in response to local in melt compositions.": should delete the second "in".
- Line 351: "crystal entertainment" should be "crystal entrainment".
- Lines 487-488 and also data points in Fig.6: what is the uncertainty range of the diffusivities estimated from experimental products from this study?

Reviewer #3:

Remarks to the Author:

See attached review doc and annotated ms file.

Review of Manuscript NCOMMS-21-01451

Title: Mixing-induced chemical disequilibrium creates and modifies basaltic crystal cargoes

Authors: David A. Neave, Philipp Beckmann, Harald Behrens and François Holtz

Reviewed: 2/27/2021 by Tom Shea

Article background: The study examines magma mixing processes and their potential effects on erasing or preserving mineral populations and the compositional characteristics of primitive minerals/melts. Isothermal mixing experiments outline the ways in which phase equilibrium is modified and the progression of disequilibrium with time. It is concluded that without any requirements for differences in temperatures, intrusion of compositionally distinct primitive melts within existing mush-rich reservoirs can substantially modify the geochemical signature of magmas that ultimately rise and erupt.

Recommendation: The article was a pleasure to read, very well written, easy to follow and with excellent illustrations to accompany the text. I wish all submissions were as clear and nice to go through! The experiments provide a great proof of concept that differences in magma composition alone can drive disequilibrium and mineral reactions that ultimately obscure some of the original characteristics of primitive melts. A slight weakness of the study at present is that the scaling and links to mush remobilization in natural systems is a bit speculative. I detail below some aspects that could be discussed to make the connection between the experimental results and primitive magmas in Iceland more obvious. The discussion can be modified to address this concern fairly readily. I therefore recommend minor revisions.

Main comments:

Extrapolating the experimental results to a natural system: The experiments presented show beautifully the progression of a diffusive mineral dissolution 'front', where diffusive depletion in Al_2O_3 is responsible for destabilization of plagioclase. It appears like the mechanism is stated in the discussion to be critical in controlling magma homogenization rates after mixing. In the discussion portion of the ms, the extrapolation of these results to a natural system consisting of primitive recharge entering a resident mush is currently a bit of a jump and requires some further justification and scaling analysis. It would be fruitful to have some more discussion and linking of the experiments and nature in the last section of the discussion. Below is a list of similarities and differences that could help guide a more detailed examination and comparison.

In particular, the experiments and natural system may be similar in:

- (1) melt compositions that were carefully chosen by the authors (although I note they differ in MgO and thus likely in 'primitivity').
- (2) Mineral compositions (good match between expe and nature in Fig. 7)
- (3) the temperatures (although they may be lower than in nature depending on how primitive the magmas are...this would only act to hasten the kinetics of the processes discussed by the authors so no problem there)

The experiments and natural Icelandic magmatic systems discussed may however differ in:

- (1) Diffusion vs. advection in mixing. There, I think the importance of advective mixing is understated. Advective mingling and mixing will, I suspect, be the primary mechanism for melt homogenization in these low viscosity magmas. This can be tested by calculating the diffusion vs. advection timescales for given thicknesses of magma (or magma filaments). See for instance the analysis in Rossi et al. (2017). The authors could compare these timescales for advective mixing of their two magmas - with the viscosities they already calculated for the Eyring diffusivities- versus diffusive homogenization. There is an important contrast in the moving diffusion front mechanism

advocated to trigger dissolution and fig. 9, which seems to show advective mush remobilization. With these added calculations, I think that the authors may have better arguments to link their diffusion-only experiments with a natural system that likely involves both homogenization mechanisms. The advective mixing equation is an imperfect model for intrusion of a melt into a possibly cohesive mush system as envisioned by the authors. But it should give some mixing time bounds with the diffusion-only scenario.

- (2) The size of the crystals that are forming and dissolving (~ order of magnitude smaller in the experiments) and their compositional complexity.
- (3) The relative fractions of the two magmas (50% each in the expts, likely much more variable in nature). This seems like an important variable that could be at least briefly discussed. What is the minimum mixing fraction for these two end-members that would ultimately lead to dissolution of part of the plagioclase cargo? That means attaining < 16 wt.% by mixing a minimum volume of 25% of the 14.5 wt.% end-member and 75% of the 16.5 wt.% Al₂O₃ magma.
- (4) The third mushy end-member. In my mind, the experiments simulate mixing of two melts of different compositions with very small crystal fractions, but resorbing the crystal part of the mush in nature makes this a third chemical component; i.e., at its bottom the resident mush-rich and Al₂O₃-rich magma will have a different 'bulk composition' than the Al₂O₃-rich melt at the top (which is the melt examined in the experiments). Disaggregating and dissolving a plag rich mush will likely enrich the mixed melt quickly in Al₂O₃ so that one might expect recrystallization? Perhaps again a question of intruded volume.

Is compositional disequilibrium a new mineral reaction mechanism? The possibility of inducing mineral reactions by changing melt compositions (rather than just temperature) is presented as an important new mechanism in the paper. It may be a bit of an overstatement, as this hypothesis is a fairly classic phase equilibrium problem ('constitutional undercooling') and has been investigated before numerically and based on simple phase equilibria arguments (in particular, see L'Heureux 1993 Phys Rev. E; Liang 2003 G-cubed, doi:10.1029/2002GC000375) as well as experimentally in the context of convective+diffusive dissolution (see Chen & Zhang 2008, 2009, Yu et al. 2016) or disaggregation and dissolution (Ruprecht et al. 2020, GRL).

Recommendation: I would simply make sure these various prior works are referenced in the introduction in the context of isothermal mineral melt reaction experiments (several already are, the Chen and Yu papers).

Why is there variation in the plag compositions? Fig. 7 shows well the degree of compositional variability in each end-member magma (around 1 Fo unit in mol%, around 2.5 Mg# for cpx, >4 An units in mol%). For olivine, the variations are close to the 2sigma but not for plag and cpx. For cpx, sector zoning is mentioned as provably responsible. Could you provide a brief explanation/hypothesis as to what may be causing these variations for plag? Do you think it may be kinetics (somehow variable crystal growth rates throughout the charge causing slight disequilibrium partitioning)?

We have addressed comments in blue below; excerpts from the revised manuscript are presented in orange with references in author date format for clarity. Responses to comments on annotated manuscripts have been uploaded separately.

Reviewer #1 (Remarks to the Author):

NCOMMS-21-01451

Dear authors and editors,

Thank you for giving me the opportunity to evaluate such work. I enjoyed the manuscript and paid a lot of attention to all aspects covering my research area. I hope my comments will be perceived as constructive as possible.

The manuscript deals about isothermal mixing of chemically different basalts, with an experimental approach, and finds applications in magma remobilisation (although that term was not used by authors) in oceanic domains.

The manuscript is well constructed, illustrated, and written. What a pleasure to evaluate a mature work! It respects NCOMMS policies in terms of length, illustrations. The availability of all data as supplementary materials is also very appreciated.

I structured my review with major/minor comments, and other minor comments directly in the pdf file. I think the manuscript is very close to be ready for publication; there is one weakness to me (major comment 4) that may drive my final decision, according to the authors' answer.

We thank the reviewer for their supportive comments and the care with which they evaluated our work. We have incorporated almost all of their comments (documented here and in the attached pdf). We were very much aware of both their work and the various studies that have emanated from LMU and Perugia on magma mixing, but did not initially cite them as they typically consider andesite genesis by basalt–dacite/rhyolite hybridisation or mixing between alkalic metals. This was an error, and we now incorporate as many previous works as possible while keeping the total number of citations as close to 70. We hope this helps our submission to have as broad an impact on the magma mixing community as possible.

Major points

1. Despite the focus on chemical mixing of this study, a textural description of mixing experiments is lacking (Related to comment L.126). Olivine concentration decreases with experiment duration, therefore, MgO & FeO concentrations in the melt increase, and relatively, Al₂O₃ concentration (directly related to Plagioclase crystallization) decreases. Please, add a (short?) textural description of the products (additional figure with zoom of end-members at 96h?) that will be connected to the discussion about the chemistry of the system.

We thank the reviewer for this very helpful comment and believe that the changes documented below have increased our manuscript's rigour considerably. By removing brief methodological descriptions at the start of the Results we have saved enough space to add more detailed textural descriptions, focussing particularly on textural evolution (or rather lack thereof) as a function of increasing experimental duration. We accompany these descriptions with a new figure (new Fig. 4) showing zoomed-in BSE images of the products of magma-magma reaction experiments away from original magma-magma interfaces. We observe that only clinopyroxene crystal show evidence for possible textural evolution (Ostwald ripening) in portions of experimental products derived from the Stapafell analogue.

We now note that olivine and clinopyroxene abundances appear to decrease with increasing duration (in the Stapafell analogue), but also discuss, with evidence from mass balance calculations, that these differences are probably the result of gravitational settling in synthesis experiments rather than textural maturation with increasing experimental duration. We also note that melt fractions and melt MgO contents are slightly higher in magma-magma reaction experiments than synthesis experiments, consistent with the latter being sampled from capsule ends just outside the hot zone of the pressure vessel used; differences in MgO contents cannot be attributed to differences in olivine contents

We have improved the accuracy of our comments about FeO* contents. Specifically, we use mass balance calculations to describe how the products of 1- and 4-hour experiments gained a small amount of Fe (a few % relative) and 24- and 96-hour experimental lost a small amount of Fe (again a few % relative). Unfortunately, some Fe exchange with capsules was inevitable with the experimental setup used, despite the efforts taken to mitigate it. Extents of Fe gain and loss were similar in both analogues, indicating that sigmoidal FeO* profiles produced after 1 and 4 hours can still be modelled (in the interests of saving space we do not discuss these final points in the manuscript.)

The products of magma-magma reaction experiments share textural and mineralogical characteristics with products of magma synthesis experiments (Figs. 3, 4). Namely, crystals are smaller and more abundant in products derived from the incompatible element-depleted Háleyjabunga analogue than those from the incompatible element-enriched Stapafell analogue. Overall, textures of crystals far from original interfaces do not appear to evolve with increasing experimental duration; only clinopyroxene crystals in portions of experimental products derived from the Stapafell analogue show possible evidence of Ostwald ripening (Fig. 4d). Although crystal fractions remain broadly constant with increasing experimental duration in far-field portions of experimental products derived from the Háleyjabunga analogue away from initial magma-magma interfaces (Figs. 3, 4a, 4b), they appear to decrease in those derived from the Stapafell analogue (Figs. 3, 4c, 4d). However, mass balance demonstrates that far-field melt mass fractions change little through time: between 1 and 96 hours, F changes from 0.77 to 0.80 in far-field portions of the Háleyjabunga analogue, and 0.92 to 0.93 in far-field portions of the Stapafell analogue. It is therefore likely that apparent differences in observed crystal fractions are related to heterogeneities created by gravitational settling during the synthesis experiment on the Stapafell analogue and that disequilibrium crystal fractions were retained through magma-magma reaction experiments. Somewhat lower crystal fractions in the products of magma-magma reaction experiments with respect to the products of synthesis experiments reflect samples of the latter being sourced from capsule ends that lay slightly outside the vessel's hot zone.

Al₂O₃ TiO₂ and K₂O profiles have sigmoidal forms that show progressively more gradual transitions between far-field compositions with increasing experimental durations (Figs. 5a, 5c, 5e). FeO profiles are sigmoidal in the products of 1- and 4-hour experiments, but near-linear in the products of 24- and 96-hour experiments (Fig. 5b). FeO* contents are also displaced to lower mean FeO* contents in the products of the 24- and 96-hour experiments with respect those of the 1- and 4-hour experiments. Mass balance indicates that this results from modest Fe loss from Au₈₀Pd₂₀ capsules in the case of former (1–2 % relative) and modest Fe gain in the case of the latter (2–5 wt.% relative)(Barr and Grove, 2010). CaO and Na₂O profiles are sigmoidal in the products of 1- and 4-hour experiments, but linear in the products of 24- and 96-hour experiments, (Figs. 5d, 5f).*

Did you notice the shape of the interface? e.g. 4h experiment with a triangle/conic shape.

We did and now note explicitly that this reflects advection during the initial establishment of interfaces that subsequently evolved by diffusion.

Importantly, the similarity of parallel profiles demonstrates that advection was negligible once magma-magma interfaces were established (early advection is reflected in curved magma-magma interfaces in the products of 1- and 4-hour experiments) and that experimental products record dominantly diffusive signals.

I am also thinking about providing, if possible, an image showing the initial texture of syntheses, even if it is very close to 1h magma-magma interaction experiment (as supplementary figure?).

We agree. At the time of submission, we did not have high-quality BSE images of synthesis experiments. However, we have since been able to collect new images and have integrated these into a revised version of Fig. 2. These new images also resolve issues with intensity gradients on some images that resulted from a faulty BSE detector.

It is important to appreciate the mineral phase proportion through experiment duration. Since you have crystal and melt chemistry, you may check if your system remains chemically closed by mass balance, and perhaps you will get some clues about iron behaviour.

We agree that these are very relevant points and thank the reviewer for making clear and specific suggestions of how to address them. Please see our extended response above where we address phase proportion evolution and Fe exchange with capsules.

2. I wish to see some clarifications about plagioclase behaviour. I have doubts about the resorption of the plagioclase shown in Fig. 8. First because some of some of its faces remains crystalline (it is subeuhedral, like other in top right corner of the same figure close to the white arrow), and also because of the presence of euhedral plagioclase crystal in the same experimental product, visible on the attached file equivalent to Fig 3d at various distances from the interface. Similar plagioclase crystals are present in the four presented experiments. Some of those euhedral crystals contain embayments or melt inclusions, but the latter could result from a relatively fast crystal growth (works of Kirkpatrick 1975, Donaldson et al. 1976, Faure et al. 2003-2005, Laumonier et al. 2019 CMP). Could you please discuss this point and strengthen your argumentation?

We thank the reviewer for directing us to look at these textures in closer detail. On reflection we agree that skeletal plagioclase crystals occur throughout most of the experimental products where plagioclase is stable – this is especially clear when comparing the new Fig. 9 (old Fig.8) with the new Fig.4. We agree with the reviewer's assessment that these textures likely reflect rapid crystal growth during synthesis experiments and have been largely unaltered during subsequent magma-magma reaction experiments. We now highlight this in the manuscript text, citing the classic work of Kirkpatrick (1975) in the Results. We agree that the other papers mentioned are certainly of relevance, but we do not have enough room to devote any more citations to this topic.

Although XAn variability exceeds analytical uncertainty in the products of all magma-magma reaction experiments (Fig. 8b), likely because of rapid crystal growth that is also reflected in skeletal textures (Figs. 2, 4)(Kirkpatrick, 1975), mean XAn does not vary systematically as a function of either position or experimental duration.

We have also edited new Fig. 9 (old Fig. 8) by removing the misleading label to a supposedly resorbing plagioclase. Instead we now highlight a cluster of anhedral plagioclase remnants that show much clear evidence for resorption. We have also re-written the

discussion describing the textures in new Fig. 9 (old Fig. 8). Specifically, we now make a distinction between euhedral to subhedral and often skeletal plagioclase crystals in the majority of experimental products and anhedral and cusped crystals at the limits of plagioclase stability. We believe that these anhedral and cusped textures are evident in the new Fig. 9, and provide more appropriate evidence for having caught resorption in action.

While plagioclase crystals are typically euhedral to subhedral and often skeletal in experimental products derived from the Háleyjabunga analogue and unaffected by diffusion within the melt (Figs. 2a, 4a, 4b), those at the limits of plagioclase stability in the products of magma-magma reaction experiments are typically anhedral and cusped (Fig. 9).

Related to this point (and L293-295), the size of plagioclase crystals in the initial syntheses may give valuable information (I would like to see images with lower magnification of the syntheses). Do you think Ostwald ripening can affect plagioclase population? (see for instance Cabane et al., 2005).

As described in our response to major comment 1, we have added new BSE images of synthesis and magma-magma reaction experiments at both high and low resolutions (new Figs. 2 and 4). These images demonstrate that there is no noticeable evolution of plagioclase textures as a function of increasing experimental duration. We agree that detailed textural investigations of the nature described by Cabane et al. may reveal finer structures and potentially evidence for Ostwald ripening as described by Cabane et al. (2005). However, considerable additional work would be required to fully explore this point, without, we believe, leading to any substantial changes to the resulting manuscript. Moreover, we do not have the space required to provide a detailed evaluation of plagioclase textures and choose instead to retain our current focus on the potential implications of magma mixing rather than the evolution of mineral textures alone.

3. To be completely convinced about the evolution of olivine phase, I need some extra arguments. First, I questioned the solid-state diffusive re-equilibration: what are the arguments to reach this conclusion? I would agree, for instance, if several chemical profiles in olivine of different experimental durations were presented that show time-dependent reaction (from the shape of the profile, as you did for melt in Fig 4). You could estimate Fe/Mg diffusion in olivine and compare with literature (partly done, L 303). Although I have no quantitative argument, I suggest the profiles presented in supplementary Fig. 1 result from the melt composition that varies in the course of the crystallization during syntheses experiments. I have access to a small area of each synthesis with Fig. 2 (that's why an image of syntheses at lower resolution would be helpful), but there is no clue of crystal zoning from SEM image, contrary to some olivines in the 24h experiment.

We thank the reviewer for prompting us to consider these observations more carefully. In addition to mediating the language we have used to describe the possible diffusive re-equilibration of olivine throughout; we have revised the manuscript text in both the Results and Discussion as well as overhauling Supplementary Fig. 1. We now highlight that some olivines in the products of longer-duration experiments attain higher X_{Fo} contents than observed in either synthesis or shorter-duration experiments as a result of diffusive re-equilibration in response to Fe loss to experiment capsules.

Some olivine crystals from longer duration experiments show evidence of diffusive re-equilibration in response to Fe loss (higher X_{Fo} contents than in the products of synthesis or shorter duration experiments) and diffusive homogenisation of melt FeO contents (X_{Fo} variance decreases from 2.9×10^{-5} after 1 hour to 9.6×10^{-6} after 96 hours). Indeed,*

olivine crystals in the products of the 24-hour experiment appear to show diffusion profiles that are absent from the products of shorter experiments (Supplementary Fig. 1).

Following the reviewer's suggestion, we have now added olivine composition profiles from the products of 1- and 4-hour experiments to Supplementary Fig.1 alongside those from the products of the 24-hour experiment. Importantly, these profiles from shorter duration experiments do not show any evidence for diffusive re-equilibration, suggesting that the zoning observed in the products of 24-hour experiments cannot have formed during the course of synthesis experiments as (entirely reasonably) suggested by the reviewer.

Although we do not have the space to engage in a detailed evaluation of intra-crystal diffusion, we have strengthened our discussion by improving the precision of our language and quantifying an illustrative characteristic diffusion lengthscale that is broadly comparable with a characteristic diffusion lengthscale calculated from Dohmen and Chakraborty (2007).

Indeed, the partial re-equilibration of olivine crystals over lengthscales of ~10 μm within 24 hours is broadly consistent with a characteristic diffusion lengthscale (i.e. x , where $x \propto \sqrt{Dt}$) (Zhang, 2010) of ~5 μm estimated using an Mg–Fe interdiffusion coefficient after Dohmen and Chakraborty (Dohmen and Chakraborty, 2007), and provides independent validation of mixing timescales estimated from diffusion profiles in natural systems (Mutch et al., 2019).

4. Paragraph L333: Main cause of crystal mush disaggregation of oceanic basalts. This paragraph (+ consequences and the last paragraph) appears to me as the kernel of the study as it will drive my opinion about whether or not this contribution is relevant to a large audience reached by NCOMMS. In brief, the authors show very well the consequence of isothermal mixing from experiments. The last obstacle to me is to demonstrate if this process is an alternative to classical reheating of magma reservoir potentially leading to crystal mush disaggregation or if it the main process occurring in oceanic magma reservoirs. In the latter case, the message is original. However, the demonstration of the importance of such process is not strong enough to me. The demonstration holds to the sentence L341: "We thus suggest that chemical rather than thermal disequilibrium may play an important role in triggering crystal mush disaggregation in some magma reservoirs " and L396 sums up my feeling:

"Overall, isothermal mixing between chemically variable primitive magmas likely plays an equally important role in generating the texturally and chemically complex crystal cargoes we observe at the surface as more widely recognised processes invoking mixing between variably evolved and thermally distinct magmas."

Consequently, to me, the title of the study should mention that last point.

We thank the reviewer for highlighting these weaknesses in our original submission. In response to this comment and a similar but more extensive comment from Reviewer #3 we have substantially revised relevant parts of the Discussion. While we have provided a detailed discussions of our changes (e.g., closer integration of experimental and natural observations, discussion of scaling considerations and more comprehensive referencing) in our response to the first main comment of Reviewer #3, we would nonetheless like to stress that we are not advocating that our proposed mechanism of mush disaggregation by chemical processes should supplant classical reheating models in all instances. Instead we aim to make a case that the necessary conditions for our model are widely met and consistent with abundant evidence of primitive mush crystals often being entrained by cool, evolved and relatively enriched melts rather than hot and primitive melts (e.g., Halldorsson et al., 2008; Lange et al., 2013; Neave et al. 2014).

We believe that a key strength of our manuscript lies in its presentation of a previously unrecognised but potentially widespread mechanism of mush disaggregation that is both consistent with diverse observations of magma reservoir behaviour and a logical consequence of mantle-derived primary melt variability. Whether our proposed mechanism dominates over recharge by hot melts (whose occurrence we do not deny) is impossible to say with the data available (our or others'). However, we do not believe that this necessarily has to lead to reduction in our manuscript's overall impact and relevance

Minor points

1. Fo content of olivine, L192 "Steps in X_{Fo} also occur in the products of 24- and 96-hour experiments". Actually, I do not agree. To me, there is insufficient data close to the interface to conclude. One could even draw a line from the data point at x=-3mm to the ones at x=3.5mm (24h experiments, Fig.7) that follows the olivine Fo distribution. Can you argue better to support your opinion, or be more careful if you agree with me?

We agree with the reviewer that our original comments were not well supported by the data presented. We have therefore reworded this section completely.

This difference is reflected in steps in mean X_{Fo} across magma-magma interfaces in the products of 1- and 4-hour experiments (Fig. 8a). Unfortunately, the abundance of olivine crystals is insufficient to determine whether these X_{Fo} steps persist in the products of 24- and 96-hour experiments.

2. From L323 to the end of the paragraph. "liquids in equilibrium with relatively evolved plagioclase crystals ($X_{An} \leq 0.7$) could not resorb grain boundaries in mushes dominated by primitive plagioclase crystals ($X_{An} > 0.8$) formed during earlier phases of higher temperature crystallisation" (L325-328). Why not? this seems counterintuitive to me; the higher the chemical potential, the faster the chemical reaction. I would rather interpret glomerocrysts as colder and solidified (i.e. cohesive, resistant) enclaves such as the so common mafic enclaves found in evolved magmas (although the temperature difference between mafic magma and more felsic host may be higher than for the case studied here). Can you nuance your opinion or provide additional argumentation, reference to support it?

We thank the reviewer for highlighting this point, which also touches on topics raised by Reviewer #3. The key part of our argument is that evolved melts in equilibrium with plagioclase will not resorb plagioclase crystals formed at higher temperatures along the same liquid line of descent but rather trigger rim crystallisation. If the melts were from different liquid lines of descent, however, then chemical potentials could be much higher, especially if plagioclase is not stable in the evolved melt, meaning that resorption could occur. We have now sought to clarify our arguments on this topic

For example, primitive crystal cargoes are generally too refractory to be chemically excavated from cohesive mushes by the relatively evolved liquids that often carry them to the surface (Lange et al., 2013a; Neave et al., 2013, 2014). That is, assuming evolution along a single liquid line of descent, the injection of liquids in equilibrium with relatively evolved plagioclase crystals ($X_{An} \leq 0.7$) would not unlock mushes of primitive plagioclase crystals ($X_{An} > 0.8$) but rather lock them up further by triggering the crystallisation of overgrowth rims.

3. L 352. I do not agree with this sentence and ask you to strengthen the argumentation. To me, crystal-melt disequilibria evidence magma mingling/mixing, but not that chemical variability plays a widespread role in driving crystal mush disaggregation.

Same for L358: “Nevertheless, the occurrence...”

We thank the reviewer for highlighting this clumsy wording that clearly failed to articulate the point we were trying to make. We agree that the key inference from the widespread occurrence of crystal-melt disequilibrium is that magma mixing is also widespread. We agree that the presence of disequilibrium alone does not provide proof that chemical variability plays a widespread role in driving mush disaggregation. Rather our intention was to argue that the presence of widespread disequilibrium means that the necessary conditions for mush disaggregation to be driven by chemical variability are similarly widespread. This is a nuanced but important point. We have therefore re-written whole paragraph to argue that A creates the necessary conditions for B to occur rather than that A proves B. We have also removed some of the details about different ridge settings, as we now believe that they served as an unnecessary distraction from the main point we were trying to make.

Chemical and isotopic disequilibria between basaltic melts and mush-derived high-XAn plagioclase crystals are well documented in oceanic basalts from ocean islands and mid-ocean ridges of all spreading rates (Ridley et al., 2006; Halldórsson et al., 2008; Costa et al., 2010; Lange et al., 2013a; Neave et al., 2014; Bennett et al., 2019), demonstrating that mixing between chemically variable primitive magmas is widespread in the oceanic realm (MacLennan, 2008a; Shorttle, 2015). Although pervasive evidence of mixing and disequilibrium does not in itself constrain the mechanisms by which mush disaggregation and crystal entrainment occur, it does highlight that the deep chemical variability required to trigger mush disaggregation by mixing-induced chemical disequilibrium and crystal resorption is globally present. Moreover, we speculate that variations in the H₂O content of arc magmas may have analogous effects to variations in the major element content of oceanic basalts, whereby the injection of H₂O-rich magmas may facilitate mush disaggregation by triggering plagioclase resorption (Sisson and Grove, 1993).

4. As pointed out also in the abstract, please describe “oceanic basalts” in the discussion and the different areas of applicability of your study.

We have removed references to oceanic basalt from the abstract for clarity, stating instead that we are simply looking at basaltic crystal cargoes. We do however provide a full definition in the first sentence of the introduction:

Chemical variability in primitive mid-ocean ridge and ocean island basalts (MORB and OIB, respectively; oceanic basalts, collectively) results from variability in mantle melting processes and source compositions (Gast, 1968; Hofmann and White, 1982).

5. What are the yellowish (saturated) “objects” in some figures? Oxides? Are they present in the syntheses? e.g. in the 24h experiment, close to the interface, there is an olivine with 2 melt inclusions and another yellow inclusion/area. Another one, with square shape in the 4h experiment, at ~1/3 from the bottom of the sample, left side, and more frequently in the same experiment, few 10 µm above the interface.

These yellow flecks are small fragments of AuPd that detached from the capsules during capsule production or sample preparation (i.e. polishing). To enhance the visibility of variations within our samples we applied a black mask to the otherwise high-BSE-intensity (i.e.) capsule materials shown in Fig 3. Otherwise, these bright spots have no geological significance, and we therefore do not draw any further attention to them. We now state this clearly in the caption to Fig. 3.

Occasional bright flecks are small fragments of capsule material detached during sample preparation and are of no geological significance.

Best regards,

Mickael Laumonier

2021 February 3rd

Annotated pdf

We thank the reviewer for their detailed reading of our submission. We have made a number of minor to moderate edits to the manuscript based on the comments in the attached pdf; responses to these comments are an annotated pdf.

Reviewer #2 (Remarks to the Author):

In their paper entitled “Mixing-induced chemical disequilibrium creates and modifies basaltic crystal cargoes”, Neave et al. report new experimental observations of isothermal magma mixing-induced chemical disequilibrium in basaltic melts and how such mixing can create and modify crystal cargoes. The authors have performed magma-magma reaction experiments at 300 MPa and 1190 °C on synthetic analogues of two types of Icelandic lavas (with distinct major element compositions) for several durations (from 1 to 96 hours), and then performed careful analysis of the experimental products. They found that the melt (near interface) experienced diffusive re-equilibration with elemental diffusivities overlapping with those reported in the literature, and different minerals reacted in different ways to the isothermal mixing: most olivines survived through diffusive re-equilibration while clinopyroxene and plagioclase had partial and complete resorption respectively after a few hours.

These results highlight the potentially significant yet overlooked effect of isothermal mixing-induced chemical disequilibrium on crystal mush disaggregation, chemical/physical properties of magmas, and the eruption dynamics, and thus have wide implications in the studies of mantle chemistry as well as magma mixing and pre-eruptive magmatic processes in basaltic systems.

Overall, I find the conclusions and data interpretation in this paper robust and convincing. The abstract is concise, and the main text is well written with clear structure. Experimental procedures and analytical methods are documented in detail, and figures are illustrated with high quality. Considering all these factors, I am glad to recommend this paper for publication in Nature Communications (with minor revisions).

We thank the reviewer for their supportive and encouraging remarks. We also thank them for some of their seemingly minor comments that nonetheless led to substantial improvements in the rigour of our manuscript.

The only question/doubt I have in mind is whether the mineral responses observed in this study (especially recrystallisation/resorption of clinopyroxene/plagioclase) would be different under other P-T-melt/mineral-composition conditions? It would be useful to comment about this in discussion. The rest of my comments and some grammatical edits are listed below.

We originally only included one sentence in the Discussion that addressed this point, but did so rather obliquely:

For example, our magma-magma reaction experiments demonstrate how the diffusive equilibration of incompatible element-depleted and plagioclase-saturated magmas with incompatible element-enriched and plagioclase-undersaturated magmas can trigger isothermal plagioclase resorption (Figs. 4, 8). However, it is straightforward to envisage slightly different scenarios in which diffusive influxes of Al₂O₃ into incompatible element-

enriched magmas instead trigger waves of crystal nucleation in systems close to plagioclase saturation or waves of rim growth in systems that already contain plagioclase.

We have re-written and expanded this section to comment on the likely effects of varying temperature and pressure. In particular, we note that at lower temperatures, Al₂O₃ exchange may trigger plagioclase nucleation or rim growth rather than resorption.

For example, our magma-magma reaction experiments at 1190 °C demonstrate how the diffusive equilibration of incompatible element-depleted and plagioclase-saturated magmas with incompatible element-enriched and plagioclase-undersaturated magmas can trigger isothermal plagioclase resorption in the former without plagioclase nucleation in the latter (Figs. 4, 8). At lower T conditions (~1170 °C (Neave et al., 2019b)), however, isothermal Al₂O₃ diffusion may trigger plagioclase nucleation in initially plagioclase-free and incompatible element-enriched magmas rather than plagioclase resorption in initially plagioclase-bearing and incompatible element-depleted magmas because the former would be closer to plagioclase saturation than is the case in our experiments. In plagioclase-saturated but chemically variable systems at yet lower T conditions (≤1160 °C (Neave et al., 2019b)), isothermal Al₂O₃, CaO and Na₂O diffusion could produce plagioclase crystals with X_{An} zonation. Comparable differences in behaviour could also arise by mixing magmas initially equilibrated at different pressures because plagioclase stability correlates with P (Husen et al., 2016).

Line-by-line comments

Line 58-59: "...at a under isothermal contitions,..." should be "...under isothermal conditions,..."?

Corrected

Line 173-176: I am wondering the range of melt water contents that were used to determine diffusivity in previous studies, and whether this has a large influence on the difference between the diffusivities reported in the literature and your estimation.

This is a very helpful comment, and we now remark briefly that H₂O affects diffusivity.

Finally, we note that melt H₂O contents were not reported in the published data considered here, and that small variations in melt H₂O contents (~1 wt.%) affect the diffusivity of network-forming and high-field strength elements considerably (Zhang, 2010).

Line 180: not sure what this sentence means. Please correct it.

We agree that this section was confusing, so we have re-written it completely.

Estimated diffusivities of Al₂O₃ and TiO₂ are similar to theoretical Eyring diffusivities ($D_{E} = (k_B T) \lambda \eta$, where D_{E} is the Eyring diffusivity, k_B is Boltzmann's constant, λ is a jump distance of 0.4 nm related to the atomic spacing of silicate liquids and η is the average viscosity of the two end-member melts (12.5 Pa.s), calculated here with the model of Giordano et al. (Giordano et al., 2008) (Glasstone et al., 1941); Eyring diffusivities are typically good at describing the behaviour of relatively high-field strength network-forming cations (i.e. Si, Al and Ti) and oxygen (González-García et al., 2018).

Lines 182-183: better to report the value of viscosity used for calculation if it is not shown elsewhere in the manuscript.

We have added the relevant viscosity (12.5 Pa.s) to the text here, and also report sample specific viscosities in the Supplementary Material.

Lines 183-185: "...possibly reflecting differences in diffusion mechanisms between network-forming and network-modifying cations.": How about FeO, which can be also regarded as network-modifying component but shows diffusivities similar to the theoretical Eyring diffusivities in Fig. 6?

We thank the reviewer for highlighting this point, and agree that the dichotomy between network-forming and network-modifying cations we originally presented was not strictly correct. In line with the discussion presented by Mungall (2002) we have edited this section to instead highlight that differences in field strength may instead place a key role in determining element diffusivities.

Eyring diffusivities are typically good at describing the behaviour of relatively high-field strength network-forming cations (i.e. Si, Al and Ti) and oxygen (González-García et al., 2018). In contrast, estimated diffusivities of lower-field strength Na₂O and K₂O are higher, reflecting variable degrees of decoupling from network-forming cations (Figs. 7e, 7f) (Dingwell, 1990; Mungall, 2002).

Line 200 and also Fig. 9 caption: the equation for XAn is usually written as $XAn = Ca/(Ca+Na+K)$, unless it is defined differently in this study.

Corrected

Line 287: "...no more that..." should be "...no more than..."

Corrected

Line 324: "...too refractory to have been thermally by...": a verb is missing between "thermally" and "by".

Corrected

Line 341: "...in response to local in melt compositions.": should delete the second "in".

We have corrected the text which now reads:

response to local changes in melt compositions

Line 351: "crystal entertainment" should be "crystal entrainment".

Woops! Corrected.

Lines 487-488 and also data points in Fig.6: what is the uncertainty range of the diffusivities estimated from experimental products from this study?

We thank the reviewer for highlighting this point, which, on reflection, we are surprised does not feature as a major comment. We now report uncertainties in diffusion coefficients in Fig. 7 and the Supplementary Data. We also now describe how these uncertainties were estimated in the Methods.

Uncertainties in diffusion coefficients were estimated by fitting composition profiles that had been repeatedly resampled according to the analytical uncertainties associated with each element.

Reviewer #3 (Remarks to the Author):

Article background: The study examines magma mixing processes and their potential effects on erasing or preserving mineral populations and the compositional characteristics of primitive minerals/melts. Isothermal mixing experiments outline the ways in which phase

equilibrium is modified and the progression of disequilibrium with time. It is concluded that without any requirements for differences in temperatures, intrusion of compositionally distinct primitive melts within existing mush-rich reservoirs can substantially modify the geochemical signature of magmas that ultimately rise and erupt.

Recommendation: The article was a pleasure to read, very well written, easy to follow and with excellent illustrations to accompany the text. I wish all submissions were as clear and nice to go through! The experiments provide a great proof of concept that differences in magma composition alone can drive disequilibrium and mineral reactions that ultimately obscure some of the original characteristics of primitive melts. A slight weakness of the study at present is that the scaling and links to mush remobilization in natural systems is a bit speculative. I detail below some aspects that could be discussed to make the connection between the experimental results and primitive magmas in Iceland more obvious. The discussion can be modified to address this concern fairly readily. I therefore recommend minor revisions.

We thank the reviewer for their detailed and careful reading of our submission and are very glad that our efforts to present a clear and digestible narrative appear to have paid off. We note that the comments below align very closely with some comments made by Reviewer #1 and we believe that by addressing them we have made our manuscript considerably more robust. We are particularly grateful for to the reviewer for their balanced and constructive approach to the review that helped us to address our manuscript weaknesses while making sure to retain its strengths.

Main comments

Extrapolating the experimental results to a natural system: The experiments presented beautifully show the progression of a diffusive mineral dissolution 'front', where diffusive depletion in Al₂O₃ is responsible for destabilization of plagioclase. It appears like the mechanism is stated in the discussion to be critical in controlling magma homogenization rates after mixing. In the discussion portion of the ms, the extrapolation of these results to a natural system consisting of primitive recharge entering a resident mush is currently a bit of a jump and requires some further justification and scaling analysis. It would be fruitful to have some more discussion and linking of the experiments and nature in the last section of the discussion. Below is a list of similarities and differences that could help guide a more detailed examination and comparison.

We thank the reviewer for this helpful guidance that has helped us to produce a much more rigorously argued manuscript. We note that this comment is very similar to Reviewer #1's fourth major comment, stressing the need to comprehensively rework the Discussion.

We have expanded the original second paragraph of the sub-section titled 'Magma mixing creates and modifies basaltic crystal cargoes' into two paragraphs. The first paragraph now aims to provide a more explicit link between our experiments on synthetic but naturalistic magma analogues with natural systems. We do this by first explaining how the Háleyjabunga analogue broadly represents plagioclase-rich mushes formed from incompatible element-enriched magmas at depth while the Stapafell analogue may reflect incompatible element-enriched recharge magmas.

Our magma-magma reaction experiments on chemically variable but naturalistic analogues of erupted Icelandic lavas suggest that injecting incompatible element-enriched and plagioclase-undersaturated magmas (like the Stapafell lava analogue) into high-XAn plagioclase-rich mushes (like the Háleyjabunga lava analogue but with a higher crystal fraction) could trigger mush disaggregation without requiring hot recharge.

The second paragraph addresses a range of potential limitations discussed below.

In particular, the experiments and natural system may be similar in:

- (1) melt compositions that were carefully chosen by the authors (although I note they differ in MgO and thus likely in 'primitivity').
- (2) Mineral compositions (good match between experiment and nature in Fig. 7)
- (3) the temperatures (although they may be lower than in nature depending on how primitive the magmas are...this would only act to hasten the kinetics of the processes discussed by the authors so no problem there)

We now explicitly emphasise the closeness of our synthetic systems to natural Icelandic systems to build a closer bridge between our targeted experimental observations and their implications for wider processes in natural systems. In the interests of keeping within the word limit we do not highlight all of these similarities more completely, but rather note that they have already been touched on in the introduction.

The experiments and natural Icelandic magmatic systems discussed may however differ in:

- (1) Diffusion vs. advection in mixing. There, I think the importance of advective mixing is understated. Advective mingling and mixing will, I suspect, be the primary mechanism for melt homogenization in these low viscosity magmas. This can be tested by calculating the diffusion vs. advection timescales for given thicknesses of magma (or magma filaments). See for instance the analysis in Rossi et al. (2017). The authors could compare these timescales for advective mixing of their two magmas - with the viscosities they already calculated for the Eyring diffusivities - versus diffusive homogenization. There is an important contrast in the moving diffusion front mechanism advocated to trigger dissolution and fig. 9, which seems to show advective mush remobilization. With these added calculations, I think that the authors may have better arguments to link their diffusion-only experiments with a natural system that likely involves both homogenization mechanisms. The advective mixing equation is an imperfect model for intrusion of a melt into a possibly cohesive mush system as envisioned by the authors. But it should give some mixing time bounds with the diffusion-only scenario.

This is an important point, and one that we did not address properly in the original submission. We have re-written completely re-written part of the Discussion as a result. We now state that advection is like to drive mush melt replacement on the macro scale (>1 mm), though we also note that diffusion is likely to play important role on the micro scale of grain boundaries where crystal-melt reactions occur. We thus hope that the Discussion is now more clearly aligned with our original aim of highlighting the importance of chemical rather than thermal disequilibrium, without having to dwell too much on the exact mechanism by which changes in melt compositions occur. We have also added citations to Laumonier et al. (2014), Rossi et al. (2017) and Ruprecht et al. (2020) to better integrate our discussions with existing literature on related topics. Rossi et al. (2017) is very relevant to our manuscript, and the idea of comparing illustrative timescales of advective and diffusive mixing is certainly appealing. However, the deceptively simple expressions of Rossi et al. (2017) incorporate a characteristic velocity related to the velocity of the mixing rods in their dynamic experiments that we cannot justifiably estimate for the systems we consider, and we thus cannot provide the simple comparison suggested by the reviewer – there are too many unknowns to make such calculations tractable.

While advection almost certainly drives magma mixing and mush melt replacement on the macro scale (>1 mm), diffusion is vital for changing melt compositions on the micro scale (<1 mm) of crystal-melt interactions that are ultimately responsible for determining crystal textures and controlling mush cohesion (Laumonier et al., 2014; Rossi et al., 2017).

(2) The size of the crystals that are forming and dissolving (~ order of magnitude smaller in the experiments) and their compositional complexity.

This is another important point, and we now note that the grain size of our experimental products is considerably smaller than the size of natural crystals. We also note, however, that the lengthscales of resorption indeed to unlock mushes are much smaller than the dimensions of whole crystals. As noted by Ruprecht et al. (2020), who we now cite, resorption within interstices and along grain boundaries is important for enabling disaggregation rather than the resorption of large whole crystals.

Although the lengthscales of plagioclase resorption we observe are short (~10 μm), partly reflecting the small size of plagioclase crystals in our experimental products with respect to natural samples (~10–20 versus >100 μm long), even modest amounts of resorption within interstices and along grain boundaries could trigger disaggregation (Ruprecht et al., 2020); large crystals do not need to resorb completely for mush cohesion to be lost, they merely need to become sufficiently detached to be entrained by their surrounding melts. That only modest amounts of resorption could enable mush disaggregation also bears on the relative volumes of different melts required to facilitate entrainment.

(3) The relative fractions of the two magmas (50% each in the experiments, likely much more variable in nature). This seems like an important variable that could be at least briefly discussed. What is the minimum mixing fraction for these two end-members that would ultimately lead to dissolution of part of the plagioclase cargo? That means attaining < 16 wt.% by mixing a minimum volume of 25% of the 14.5 wt.% end-member and 75% of the 16.5 wt.% Al₂O₃ magma.

We now describe how melt Al₂O₃ contents will be sufficiently low in the products of our ~50:50 magma-magma reactions experiments at equilibrium for all plagioclase crystals to resorb. We also note that once the proportion of the enriched end-member drops below ~0.25 that plagioclase will actually be stable at equilibrium. However, until complete equilibration has been achieved, we argue that a transient wave of plagioclase resorption will still occur and feasibly trigger disaggregation, which is essentially irreversible over short timescales. We also note that the subsequent recrystallization may place a role in creating the textural complexity observed in erupted crystal cargoes.

Our lava analogues were reacted in approximately 50:50 proportions, meaning that melt Al₂O₃ contents would eventually reach sufficiently low levels throughout the experimental samples for all plagioclase crystals to resorb (i.e. below ~16.0 wt.%). Conversely, if the proportion of incompatible element-enriched lava analogue were below ~0.25 then plagioclase would be stable upon complete re-equilibration (i.e. melt Al₂O₃ contents would exceed ~16 wt.%), re-crystallising in regions where it had previously resorbed, albeit with a lower X_{An}. However, even transient resorption triggered by volumetrically minor injections of incompatible element-enriched magmas could permit the one-way process of disaggregation, with subsequent re-crystallisation feasibly contributing to the textural complexity of erupted crystals (Bennett et al., 2019; van Gerve et al., 2020).

(4) The third mushy end-member. In my mind, the experiments simulate mixing of two melts of different compositions with very small crystal fractions, but resorbing the crystal part of the mush in nature makes this a third chemical component; i.e., at its bottom the resident mush-

rich and Al₂O₃-rich magma will have a different 'bulk composition' than the Al₂O₃-rich melt at the top (which is the melt examined in the experiments). Disaggregating and dissolving a plagioclase rich mush will likely enrich the mixed melt quickly in Al₂O₃ so that one might expect recrystallization? Perhaps again a question of intruded volume.

See the response to the point above – we now note how plagioclase resorption may be transient, with recrystallisation occurring once melt compositions have more fully re-equilibrated. We thank the reviewer for this comment, which highlights that there is much more to be done at the to understand processes that straddle mush disaggregation and reactive porous flow.

See above

Is compositional disequilibrium a new mineral reaction mechanism? The possibility of inducing mineral reactions by changing melt compositions (rather than just temperature) is presented as an important new mechanism in the paper. It may be a bit of an overstatement, as this hypothesis is a fairly classic phase equilibrium problem ('constitutional undercooling') and has been investigated before numerically and based on simple phase equilibria arguments (in particular, see L'Heureux 1993 Phys Rev. E; Liang 2003 G-cubed, doi:10.1029/2002GC000375) as well as experimentally in the context of convective+diffusive dissolution (see Chen & Zhang 2008, 2009, Yu et al. 2016) or disaggregation and dissolution (Ruprecht et al. 2020, GRL). Recommendation: I would simply make sure these various prior works are referenced in the introduction in the context of isothermal mineral melt reaction experiments (several already are, the Chen and Yu papers).

We thank the reviewer for this useful comment and for directing us to these important papers that, especially in the case of Liang (2003), were overlooked in my prior reading. Both in response to this comment and similar comments from Reviewer #1, we have rewritten the Introduction paragraph on experimental simulations considerably. In addition to adding citations to Morgavi et al (2013) and Laumonier et al. (2014) on dynamic crystallisation experiments, we also now cite the key papers of L'Heureux (1993) and Liang (2003) alongside Chen and Zhang (2008, 2009) and Yu et al. (2016). We also now make explicit reference to constitutional undercooling and have mediated our language in the discussion to highlight that some of the concepts that we address are more established than our prior wording implied. We have also integrated the study of Ruprecht et al. (2020) into the Introduction and Discussion – we thank the reviewer for pointing out this work, which in many ways is the most analogous study to our own that we yet know off.

Experimental simulations of magma mixing typically focus on physical mingling in dynamic experiments (Kouchi and Sunagawa, 1983; De Campos et al., 2004; Morgavi et al., 2013; Laumonier et al., 2014; Rossi et al., 2017) or diffusive re-equilibration in classic melt-melt couples (Kress and Ghiorso, 1993; González-García et al., 2018). While physical mingling is an essential component of magma mixing, diffusion over short (i.e. μm) lengthscales ultimately changes local melt compositions and leads to the modification of crystals by resorption, (re-)crystallisation and solid-state diffusion. However, melt-melt couple experiments are, by definition, performed under superliquidus conditions and thus provide limited insights into magma-magma reactions (Kress and Ghiorso, 1993; LaTourrette et al., 1996). Although constitutional undercooling and diffusive controls over crystal growth and resorption have been investigated in some simple systems (L'Heureux, 1993; Liang, 2003; Chen and Zhang, 2008, 2009; Yu et al., 2016), and crystal-bearing mixing experiments have been performed in the contexts of enclave formation and andesite formation by magma hybridisation (Laumonier et al., 2014; Ruprecht et al., 2020), current observations typically

constrain how pressure and temperature affect mineral stabilities rather than melt composition.

Why is there variation in the plag compositions? Fig. 7 shows well the degree of compositional variability in each end-member magma (around 1 Fo unit in mol%, around 2.5 Mg# for cpx, >4 An units in mol%). For olivine, the variations are close to the 2sigma but not for plag and cpx. For cpx, sector zoning is mentioned as provably responsible. Could you provide a brief explanation/hypothesis as to what may be causing these variations for plag? Do you think it may be kinetics (somehow variable crystal growth rates throughout the charge causing slight disequilibrium partitioning)?

We agree with the reviewer that the variability we observed in plagioclase XAn contents was surprisingly large. We are anecdotally aware that such variability can be quite common in notionally equilibrium experiments, but surmise that it is rarely pick up on because of the tendency to only report average mineral compositions in phase 'equilibria' studies. As the reviewer notes, we believe that sector zoning formed during crystallisation under slightly disequilibrium conditions can result in considerable variability in notionally equilibrium clinopyroxene compositions (as documented by Neave et al., 2019, JPet). In response to this comment and a similar comment by Reviewer #1, we thus believe that similar processes occur with respect to plagioclase. Namely, we think that variability in plagioclase XAn, which is of comparable magnitude in the products of all synthesis and magma-magma reaction experiments where plagioclase is stable, is inherited from rapid crystal growth during initial plagioclase formation. We also note that this explanation of observed plagioclase chemistry is consistent with the presence of skeletal textures as described by Kirkpatrick (1975), who we now cite. Unfortunately, it only seems like that we'll make progress on issues like this as a community when full EPMA datasets are routinely published alongside mean values.

Although XAn variability exceeds analytical uncertainty in the products of all magma-magma reaction experiments (Fig. 8b), likely because of rapid crystal growth that is also reflected in skeletal textures (Figs. 2, 4)(Kirkpatrick, 1975), mean XAn does not vary systematically as a function of either position or experimental duration.

Reviewers' Comments:

Reviewer #2:

Remarks to the Author:

This study reports new experimental observations of isothermal magma mixing-induced chemical disequilibrium in basaltic melts and the modification of such mixing to crystal cargoes. The authors performed magma-magma reaction experiments on synthetic analogues of two types of Icelandic lavas with distinct major element compositions and performed careful analysis of the experimental products. Their results show that the melt experienced diffusive re-equilibration with elemental diffusivities mostly overlapping with those reported in the literature, and different minerals reacted to the mixing in different ways where most olivines survived through diffusive re-equilibration whereas clinopyroxene and plagioclase did not. The findings from this study are novel and they highlight the effect of isothermal mixing-induced chemical disequilibrium on crystal mush disaggregation, chemical/physical properties of magmas, and the eruption dynamics, and thus may provide new insights into a wide range of topics such as mantle chemistry and the pre-eruptive magmatic processes in basaltic systems.

Having re-reviewed the paper I find that the original comments I made have been addressed in a satisfactory manner, including additional discussions on the P-T effect on the crystal (plagioclase) growth/resorption, the melt water effect on elemental diffusivities, and uncertainties in the diffusivities determined from this study. The paper is very well written. The experimental procedure and analytical conditions are described in detail. The figures are well drafted and of high quality. Overall, I am glad to recommend this paper for publication in Nature Communications.

Weiran Li

June 21, 2021

Reviewer #3:

Remarks to the Author:

I read the response document as well as the revised manuscript. I applaud the authors for making excellent changes and tackling the issues raised head on. I have no other reservations and recommend the paper be accepted.

Reviewer #4:

Remarks to the Author:

Great job !

The comments of the three reviewers have been incorporated.

I recommend the publication of the manuscript in this last version.

Mickael Laumonier